# On gradient regularizers for MMD GANs

**Michael Arbel**[*]
Gatsby Computational Neuroscience Unit
University College London
michael.n.arbel@gmail.com

**Danica J. Sutherland**[*]
Gatsby Computational Neuroscience Unit
University College London
djs@djsutherland.ml

**Mikołaj Bińkowski**
Department of Mathematics
Imperial College London
mikbinkowski@gmail.com

**Arthur Gretton**
Gatsby Computational Neuroscience Unit
University College London
arthur.gretton@gmail.com

## Abstract

We propose a principled method for gradient-based regularization of the critic of GAN-like models trained by adversarially optimizing the kernel of a Maximum Mean Discrepancy (MMD). We show that controlling the gradient of the critic is vital to having a sensible loss function, and devise a method to enforce exact, analytical gradient constraints at no additional cost compared to existing approximate techniques based on additive regularizers. The new loss function is provably continuous, and experiments show that it stabilizes and accelerates training, giving image generation models that outperform state-of-the art methods on $160 \times 160$ CelebA and $64 \times 64$ unconditional ImageNet.

## 1 Introduction

There has been an explosion of interest in *implicit generative models* (IGMs) over the last few years, especially after the introduction of generative adversarial networks (GANs) [16]. These models allow approximate samples from a complex high-dimensional target distribution $\mathbb{P}$, using a model distribution $\mathbb{Q}_\theta$, where estimation of likelihoods, exact inference, and so on are not tractable. GAN-type IGMs have yielded very impressive empirical results, particularly for image generation, far beyond the quality of samples seen from most earlier generative models [e.g. 18, 22, 23, 24, 38].

These excellent results, however, have depended on adding a variety of methods of regularization and other tricks to stabilize the notoriously difficult optimization problem of GANs [38, 42]. Some of this difficulty is perhaps because when a GAN is viewed as minimizing a discrepancy $\mathcal{D}_{\text{GAN}}(\mathbb{P}, \mathbb{Q}_\theta)$, its gradient $\nabla_\theta \mathcal{D}_{\text{GAN}}(\mathbb{P}, \mathbb{Q}_\theta)$ does not provide useful signal to the generator if the target and model distributions are not absolutely continuous, as is nearly always the case [2].

An alternative set of losses are the integral probability metrics (IPMs) [36], which can give credit to models $\mathbb{Q}_\theta$ "near" to the target distribution $\mathbb{P}$ [3, 8, Section 4 of 15]. IPMs are defined in terms of a *critic function*: a "well behaved" function with large amplitude where $\mathbb{P}$ and $\mathbb{Q}_\theta$ differ most. The IPM is the difference in the expected critic under $\mathbb{P}$ and $\mathbb{Q}_\theta$, and is zero when the distributions agree. The Wasserstein IPMs, whose critics are made smooth via a Lipschitz constraint, have been particularly successful in IGMs [3, 14, 18]. But the Lipschitz constraint must hold uniformly, which can be hard to enforce. A popular approximation has been to apply a gradient constraint only in expectation [18]: the critic's gradient norm is constrained to be small on points chosen uniformly between $\mathbb{P}$ and $\mathbb{Q}$.

Another class of IPMs used as IGM losses are the Maximum Mean Discrepancies (MMDs) [17], as in [13, 28]. Here the critic function is a member of a reproducing kernel Hilbert space (except in [50], who learn a deep approximation to an RKHS critic). Better performance can be obtained,

---

[*]These authors contributed equally.

however, when the MMD kernel is not based directly on image pixels, but on learned features of images. Wasserstein-inspired gradient regularization approaches can be used on the MMD critic when learning these features: [27] uses weight clipping [3], and [5, 7] use a gradient penalty [18].

The recent Sobolev GAN [33] uses a similar constraint on the expected gradient norm, but phrases it as estimating a Sobolev IPM rather than loosely approximating Wasserstein. This expectation can be taken over the same distribution as [18], but other measures are also proposed, such as $(\mathbb{P} + \mathbb{Q}_\theta)/2$. A second recent approach, the spectrally normalized GAN [32], controls the Lipschitz constant of the critic by enforcing the spectral norms of the weight matrices to be 1. Gradient penalties also benefit GANs based on $f$-divergences [37]: for instance, the spectral normalization technique of [32] can be applied to the critic network of an $f$-GAN. Alternatively, a gradient penalty can be defined to approximate the effect of blurring $\mathbb{P}$ and $\mathbb{Q}_\theta$ with noise [40], which addresses the problem of non-overlapping support [2]. This approach has recently been shown to yield locally convergent optimization in some cases with non-continuous distributions, where the original GAN does not [30].

In this paper, we introduce a novel regularization for the MMD GAN critic of [5, 7, 27], which *directly targets generator performance*, rather than adopting regularization methods intended to approximate Wasserstein distances [3, 18]. The new MMD regularizer derives from an approach widely used in semi-supervised learning [10, Section 2], where the aim is to define a classification function $f$ which is positive on $\mathbb{P}$ (the positive class) and negative on $\mathbb{Q}_\theta$ (negative class), in the absence of labels on many of the samples. The decision boundary between the classes is assumed to be in a region of low density for both $\mathbb{P}$ and $\mathbb{Q}_\theta$: $f$ should therefore be flat where $\mathbb{P}$ and $\mathbb{Q}_\theta$ have support (areas with constant label), and have a larger slope in regions of low density. Bousquet et al. [10] propose as their regularizer on $f$ a sum of the variance and a density-weighted gradient norm.

We adopt a related penalty on the MMD critic, with the difference that we only apply the penalty on $\mathbb{P}$: thus, the critic is flatter where $\mathbb{P}$ has high mass, but does not vanish on the generator samples from $\mathbb{Q}_\theta$ (which we optimize). In excluding $\mathbb{Q}_\theta$ from the critic function constraint, we also avoid the concern raised by [32] that a critic depending on $\mathbb{Q}_\theta$ will change with the current minibatch – potentially leading to less stable learning. The resulting discrepancy is no longer an integral probability metric: it is asymmetric, and the critic function class depends on the target $\mathbb{P}$ being approximated.

We first discuss in Section 2 how MMD-based losses can be used to learn implicit generative models, and how a naive approach could fail. This motivates our new discrepancies, introduced in Section 3. Section 4 demonstrates that these losses outperform state-of-the-art models for image generation.

## 2 Learning implicit generative models with MMD-based losses

An IGM is a model $\mathbb{Q}_\theta$ which aims to approximate a target distribution $\mathbb{P}$ over a space $\mathcal{X} \subseteq \mathbb{R}^d$. We will define $\mathbb{Q}_\theta$ by a *generator* function $G_\theta : \mathcal{Z} \to \mathcal{X}$, implemented as a deep network with parameters $\theta$, where $\mathcal{Z}$ is a space of latent codes, say $\mathbb{R}^{128}$. We assume a fixed distribution on $\mathcal{Z}$, say $Z \sim \text{Uniform}\left([-1, 1]^{128}\right)$, and call $\mathbb{Q}_\theta$ the distribution of $G_\theta(Z)$. We will consider learning by minimizing a discrepancy $\mathcal{D}$ between distributions, with $\mathcal{D}(\mathbb{P}, \mathbb{Q}_\theta) \geq 0$ and $\mathcal{D}(\mathbb{P}, \mathbb{P}) = 0$, which we call our *loss*. We aim to minimize $\mathcal{D}(\mathbb{P}, \mathbb{Q}_\theta)$ with stochastic gradient descent on an estimator of $\mathcal{D}$.

In the present work, we will build losses $\mathcal{D}$ based on the Maximum Mean Discrepancy,

$$\text{MMD}_k(\mathbb{P}, \mathbb{Q}) = \sup_{f \, : \, \|f\|_{\mathcal{H}_k} \leq 1} \mathbb{E}_{X \sim \mathbb{P}}[f(X)] - \mathbb{E}_{Y \sim \mathbb{Q}}[f(Y)], \qquad (1)$$

an integral probability metric where the critic class is the unit ball within $\mathcal{H}_k$, the reproducing kernel Hilbert space with a kernel $k$. The optimization in (1) admits a simple closed-form optimal critic, $f^*(t) \propto \mathbb{E}_{X \sim \mathbb{P}}[k(X, t)] - \mathbb{E}_{Y \sim \mathbb{Q}}[k(Y, t)]$. There is also an unbiased, closed-form estimator of $\text{MMD}_k^2$ with appealing statistical properties [17] – in particular, its sample complexity is *independent* of the dimension of $\mathcal{X}$, compared to the exponential dependence [52] of the Wasserstein distance

$$\mathcal{W}(\mathbb{P}, \mathbb{Q}) = \sup_{f \, : \, \|f\|_{\text{Lip}} \leq 1} \mathbb{E}_{X \sim \mathbb{P}}[f(X)] - \mathbb{E}_{Y \sim \mathbb{Q}}[f(Y)]. \qquad (2)$$

The MMD is *continuous in the weak topology* for any bounded kernel with Lipschitz embeddings [46, Theorem 3.2(b)], meaning that if $\mathbb{P}_n$ converges in distribution to $\mathbb{P}$, $\mathbb{P}_n \xrightarrow{D} \mathbb{P}$, then $\text{MMD}(\mathbb{P}_n, \mathbb{P}) \to 0$. ($\mathcal{W}$ is continuous in the slightly stronger Wasserstein topology [51, Definition 6.9]; $\mathbb{P}_n \xrightarrow{\mathcal{W}} \mathbb{P}$ implies

$\mathbb{P}_n \xrightarrow{D} \mathbb{P}$, and the two notions coincide if $\mathcal{X}$ is bounded.) Continuity means the loss can provide better signal to the generator as $\mathbb{Q}_\theta$ approaches $\mathbb{P}$, as opposed to e.g. Jensen-Shannon where the loss could be constant until suddenly jumping to 0 [e.g. 3, Example 1]. The MMD is also *strict*, meaning it is zero iff $\mathbb{P} = \mathbb{Q}_\theta$, for *characteristic* kernels [45]. The Gaussian kernel yields an MMD both continuous in the weak topology and strict. Thus in principle, one need not conduct any alternating optimization in an IGM at all, but merely choose generator parameters $\theta$ to minimize $\mathrm{MMD}_k$.

Despite these appealing properties, using simple pixel-level kernels leads to poor generator samples [8, 13, 28, 48]. More recent MMD GANs [5, 7, 27] achieve better results by using a parameterized *family* of kernels, $\{k_\psi\}_{\psi \in \Psi}$, in the Optimized MMD loss previously studied by [44, 46]:

$$\mathcal{D}_{\mathrm{MMD}}^\Psi(\mathbb{P}, \mathbb{Q}) := \sup_{\psi \in \Psi} \mathrm{MMD}_{k_\psi}(\mathbb{P}, \mathbb{Q}). \tag{3}$$

We primarily consider kernels defined by some fixed kernel $K$ on top of a learned low-dimensional representation $\phi_\psi : \mathcal{X} \to \mathbb{R}^s$, i.e. $k_\psi(x, y) = K(\phi_\psi(x), \phi_\psi(y))$, denoted $k_\psi = K \circ \phi_\psi$. In practice, $K$ is a simple characteristic kernel, e.g. Gaussian, and $\phi_\psi$ is usually a deep network with output dimension say $s = 16$ [7] or even $s = 1$ (in our experiments). If $\phi_\psi$ is powerful enough, this choice is sufficient; we need not try to ensure each $k_\psi$ is characteristic, as did [27].

**Proposition 1.** *Suppose $k = K \circ \phi_\psi$, with $K$ characteristic and $\{\phi_\psi\}$ rich enough that for any $\mathbb{P} \neq \mathbb{Q}$, there is a $\psi \in \Psi$ for which $\phi_\psi \# \mathbb{P} \neq \phi_\psi \# \mathbb{Q}$.[2] Then if $\mathbb{P} \neq \mathbb{Q}$, $\mathcal{D}_{\mathrm{MMD}}^\Psi(\mathbb{P}, \mathbb{Q}) > 0$.*

*Proof.* Let $\hat{\psi} \in \Psi$ be such that $\phi_{\hat{\psi}}(\mathbb{P}) \neq \phi_{\hat{\psi}}(\mathbb{Q})$. Then, since $K$ is characteristic,

$$\mathcal{D}_{\mathrm{MMD}}^\Psi(\mathbb{P}, \mathbb{Q}) = \sup_{\psi \in \Psi} \mathrm{MMD}_K(\phi_\psi \# \mathbb{P}, \phi_\psi \# \mathbb{Q}) \geq \mathrm{MMD}_K(\phi_{\hat{\psi}} \# \mathbb{P}, \phi_{\hat{\psi}} \# \mathbb{Q}) > 0. \qquad \square$$

To estimate $\mathcal{D}_{\mathrm{MMD}}^\Psi$, one can conduct alternating optimization to estimate a $\hat{\psi}$ and then update the generator according to $\mathrm{MMD}_{k_{\hat{\psi}}}$, similar to the scheme used in GANs and WGANs. (This form of estimator is justified by an envelope theorem [31], although it is invariably biased [7].) Unlike $\mathcal{D}_{\mathrm{GAN}}$ or $\mathcal{W}$, fixing a $\hat{\psi}$ and optimizing the generator still yields a sensible distance $\mathrm{MMD}_{k_{\hat{\psi}}}$.

Early attempts at minimizing $\mathcal{D}_{\mathrm{MMD}}^\Psi$ in an IGM, though, were unsuccessful [48, footnote 7]. This could be because for some kernel classes, $\mathcal{D}_{\mathrm{MMD}}^\Psi$ is stronger than Wasserstein or MMD.

**Example 1** (DiracGAN [30]). *We wish to model a point mass at the origin of $\mathbb{R}$, $\mathbb{P} = \delta_0$, with any possible point mass, $\mathbb{Q}_\theta = \delta_\theta$ for $\theta \in \mathbb{R}$. We use a Gaussian kernel of any bandwidth, which can be written as $k_\psi = K \circ \phi_\psi$ with $\phi_\psi(x) = \psi x$ for $\psi \in \Psi = \mathbb{R}$ and $K(a, b) = \exp\left(-\frac{1}{2}(a - b)^2\right)$. Then*

$$\mathrm{MMD}_{k_\psi}^2(\delta_0, \delta_\theta) = 2\left(1 - \exp\left(-\frac{1}{2}\psi^2\theta^2\right)\right), \qquad \mathcal{D}_{\mathrm{MMD}}^\Psi(\delta_0, \delta_\theta) = \begin{cases} \sqrt{2} & \theta \neq 0 \\ 0 & \theta = 0 \end{cases}.$$

Considering $\mathcal{D}_{\mathrm{MMD}}^\Psi(\delta_0, \delta_{1/n}) = \sqrt{2} \not\to 0$, even though $\delta_{1/n} \xrightarrow{\mathcal{W}} \delta_0$, shows that the Optimized MMD distance is not continuous in the weak or Wasserstein topologies.

This also causes optimization issues. Figure 1 (a) shows gradient vector fields in parameter space, $v(\theta, \psi) \propto \left(-\nabla_\theta \mathrm{MMD}_{k_\psi}^2(\delta_0, \delta_\theta), \nabla_\psi \mathrm{MMD}_{k_\psi}^2(\delta_0, \delta_\theta)\right)$. Some sequences following $v$ (e.g. A) converge to an optimal solution $(0, \psi)$, but some (B) move in the wrong direction, and others (C) are stuck because there is essentially no gradient. Figure 1 (c, red) shows that the optimal $\mathcal{D}_{\mathrm{MMD}}^\Psi$ critic is very sharp near $\mathbb{P}$ and $\mathbb{Q}$; this is less true for cases where the algorithm converged.

We can avoid these issues if we ensure a bounded Lipschitz critic:[3]

**Proposition 2.** *Assume the critics $f_\psi(x) = (\mathbb{E}_{X \sim \mathbb{P}} k_\psi(X, x) - \mathbb{E}_{Y \sim \mathbb{Q}} k_\psi(Y, x)) / \mathrm{MMD}_{k_\psi}(\mathbb{P}, \mathbb{Q})$ are uniformly bounded and have a common Lipschitz constant: $\sup_{x \in \mathcal{X}, \psi \in \Psi} |f_\psi(x)| < \infty$ and $\sup_{\psi \in \Psi} \|f_\psi\|_{\mathrm{Lip}} < \infty$. In particular, this holds when $k_\psi = K \circ \phi_\psi$ and*

$$\sup_{a \in \mathbb{R}^s} K(a, a) < \infty, \quad \|K(a, \cdot) - K(b, \cdot)\|_{\mathcal{H}_K} \leq L_K \|a - b\|_{\mathbb{R}^s}, \quad \sup_{\psi \in \Psi} \|\phi_\psi\|_{\mathrm{Lip}} \leq L_\phi < \infty.$$

*Then $\mathcal{D}_{\mathrm{MMD}}^\Psi$ is continuous in the weak topology: if $\mathbb{P}_n \xrightarrow{D} \mathbb{P}$, then $\mathcal{D}_{\mathrm{MMD}}^\Psi(\mathbb{P}_n, \mathbb{P}) \to 0$.*

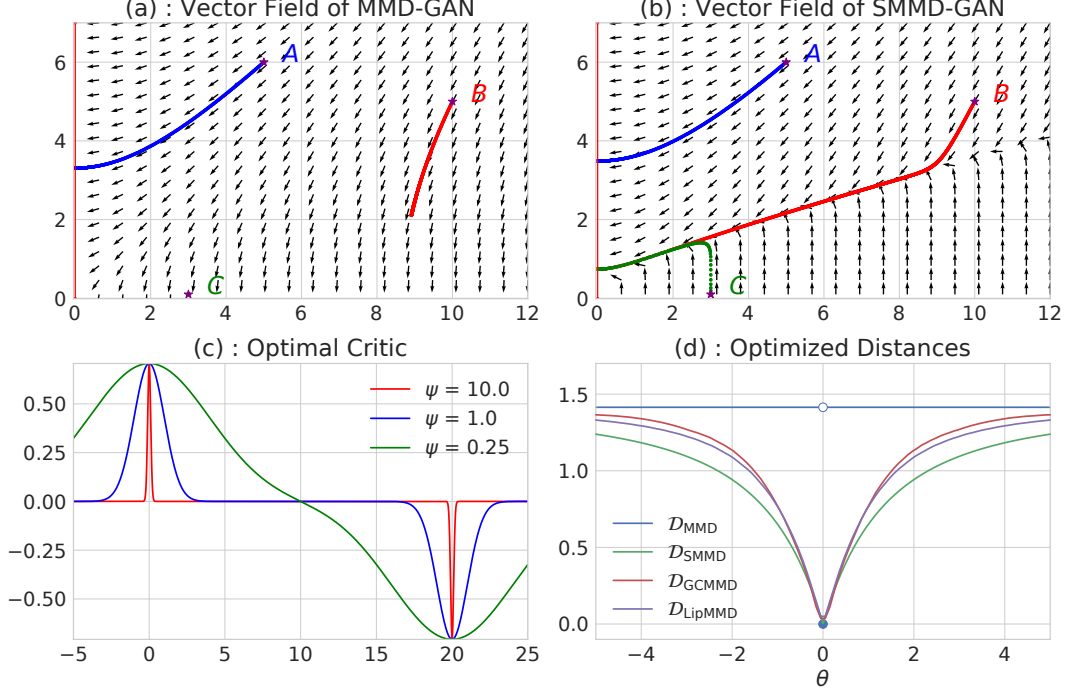

Figure 1: The setting of Example 1. (a, b): parameter-space gradient fields for the MMD and the SMMD (Section 3.3); the horizontal axis is $\theta$, and the vertical $1/\psi$. (c): optimal MMD critics for $\theta = 20$ with different kernels. (d): the MMD and the distances of Section 3 optimized over $\psi$.

*Proof.* The main result is [12, Corollary 11.3.4]. To show the claim for $k_\psi = K \circ \phi_\psi$, note that $|f_\psi(x) - f_\psi(y)| \leq \|f_\psi\|_{\mathcal{H}_{k_\psi}} \|k_\psi(x, \cdot) - k_\psi(y, \cdot)\|_{\mathcal{H}_{k_\psi}}$, which since $\|f_\psi\|_{\mathcal{H}_{k_\psi}} = 1$ is

$$\|K(\phi_\psi(x), \cdot) - K(\phi_\psi(y), \cdot)\|_{\mathcal{H}_K} \leq L_K \|\phi_\psi(x) - \phi_\psi(y)\|_{\mathbb{R}^s} \leq L_K L_\phi \|x - y\|_{\mathbb{R}^d}. \qquad \square$$

Indeed, if we put a box constraint on $\psi$ [27] or regularize the gradient of the critic function [7], the resulting MMD GAN generally matches or outperforms WGAN-based models. Unfortunately, though, an additive gradient penalty doesn't substantially change the vector field of Figure 1 (a), as shown in Figure 5 (Appendix B). We will propose distances with much better convergence behavior.

## 3 New discrepancies for learning implicit generative models

Our aim here is to introduce a discrepancy that can provide useful gradient information when used as an IGM loss. Proofs of results in this section are deferred to Appendix A.

### 3.1 Lipschitz Maximum Mean Discrepancy

Proposition 2 shows that an MMD-like discrepancy can be continuous under the weak topology even when optimizing over kernels, if we directly restrict the critic functions to be Lipschitz. We can easily define such a distance, which we call the Lipschitz MMD: for some $\lambda > 0$,

$$\mathrm{LipMMD}_{k,\lambda}(\mathbb{P}, \mathbb{Q}) := \sup_{f \in \mathcal{H}_k : \|f\|_{\mathrm{Lip}}^2 + \lambda \|f\|_{\mathcal{H}_k}^2 \leq 1} \mathbb{E}_{X \sim \mathbb{P}}[f(X)] - \mathbb{E}_{Y \sim \mathbb{Q}}[f(Y)]. \tag{4}$$

For a universal kernel $k$, we conjecture that $\lim_{\lambda \to 0} \mathrm{LipMMD}_{k,\lambda}(\mathbb{P}, \mathbb{Q}) \to \mathcal{W}(\mathbb{P}, \mathbb{Q})$. But for any $k$ and $\lambda$, LipMMD is upper-bounded by $\mathcal{W}$, as (4) optimizes over a smaller set of functions than (2). Thus $\mathcal{D}_{\mathrm{LipMMD}}^{\Psi,\lambda}(\mathbb{P}, \mathbb{Q}) := \sup_{\psi \in \Psi} \mathrm{LipMMD}_{k_\psi,\lambda}(\mathbb{P}, \mathbb{Q})$ is also upper-bounded by $\mathcal{W}$, and hence is continuous in the Wasserstein topology. It also shows excellent empirical behavior on Example 1 (Figure 1 (d), and Figure 5 in Appendix B). But estimating $\mathrm{LipMMD}_{k,\lambda}$, let alone $\mathcal{D}_{\mathrm{LipMMD}}^{\Psi,\lambda}$, is in general extremely difficult (Appendix D), as finding $\|f\|_{\mathrm{Lip}}$ requires optimization in the input space. Constraining the *mean* gradient rather than the *maximum*, as we will do next, is far more tractable.

## 3.2 Gradient-Constrained Maximum Mean Discrepancy

We define the Gradient-Constrained MMD for $\lambda > 0$ and using some measure $\mu$ as

$$\text{GCMMD}_{\mu,k,\lambda}(\mathbb{P}, \mathbb{Q}) := \sup_{f \in \mathcal{H}_k \,:\, \|f\|_{S(\mu),k,\lambda} \leq 1} \mathbb{E}_{X \sim \mathbb{P}}\left[f(X)\right] - \mathbb{E}_{Y \sim \mathbb{Q}}\left[f(Y)\right], \tag{5}$$

$$\text{where } \|f\|_{S(\mu),k,\lambda}^2 := \|f\|_{L^2(\mu)}^2 + \|\nabla f\|_{L^2(\mu)}^2 + \lambda\|f\|_{\mathcal{H}_k}^2. \tag{6}$$

$\|\cdot\|_{L^2(\mu)}^2 = \int \|\cdot\|^2 \, \mu(\mathrm{d}x)$ denotes the squared $L^2$ norm. Rather than directly constraining the Lipschitz constant, the second term $\|\nabla f\|_{L^2(\mu)}^2$ encourages the function $f$ to be flat where $\mu$ has mass. In experiments we use $\mu = \mathbb{P}$, flattening the critic near the target sample. We add the first term following [10]: in one dimension and with $\mu$ uniform, $\|\cdot\|_{S(\mu),\cdot,0}$ is then an RKHS norm with the kernel $\kappa(x, y) = \exp(-\|x - y\|)$, which is also a Sobolev space. The correspondence to a Sobolev norm is lost in higher dimensions [53, Ch. 10], but we also found the first term to be beneficial in practice.

We can exploit some properties of $\mathcal{H}_k$ to compute (5) analytically. Call the difference in kernel mean embeddings $\eta := \mathbb{E}_{X \sim \mathbb{P}}[k(X, \cdot)] - \mathbb{E}_{Y \sim \mathbb{Q}}[k(Y, \cdot)] \in \mathcal{H}_k$; recall $\text{MMD}(\mathbb{P}, \mathbb{Q}) = \|\eta\|_{\mathcal{H}_k}$.

**Proposition 3.** *Let $\hat{\mu} = \sum_{m=1}^M \delta_{X_m}$. Define $\eta(X) \in \mathbb{R}^M$ with $m$th entry $\eta(X_m)$, and $\nabla\eta(X) \in \mathbb{R}^{Md}$ with $(m, i)$th entry[4] $\partial_i\eta(X_m)$. Then under Assumptions (A) to (D) in Appendix A.1,*

$$\text{GCMMD}_{\hat{\mu},k,\lambda}^2(\mathbb{P}, \mathbb{Q}) = \frac{1}{\lambda}\left(\text{MMD}^2(\mathbb{P}, \mathbb{Q}) - \bar{P}(\eta)\right)$$

$$\bar{P}(\eta) = \begin{bmatrix} \eta(X) \\ \nabla\eta(X) \end{bmatrix}^\mathsf{T} \left(\begin{bmatrix} K & G^\mathsf{T} \\ G & H \end{bmatrix} + M\lambda I_{M+Md}\right)^{-1} \begin{bmatrix} \eta(X) \\ \nabla\eta(X) \end{bmatrix},$$

*where $K$ is the kernel matrix $K_{m,m'} = k(X_m, X_{m'})$, $G$ is the matrix of left derivatives[5] $G_{(m,i),m'} = \partial_i k(X_m, X_{m'})$, and $H$ that of derivatives of both arguments $H_{(m,i),(m',j)} = \partial_i \partial_{j+d} k(X_m, X_{m'})$.*

As long as $\mathbb{P}$ and $\mathbb{Q}$ have integrable first moments, and $\mu$ has second moments, Assumptions (A) to (D) are satisfied e.g. by a Gaussian or linear kernel on top of a differentiable $\phi_\psi$. We can thus estimate the GCMMD based on samples from $\mathbb{P}$, $\mathbb{Q}$, and $\mu$ by using the empirical mean $\hat{\eta}$ for $\eta$.

This discrepancy indeed works well in practice: Appendix F.2 shows that optimizing our estimate of $\mathcal{D}_{\text{GCMMD}}^{\mu,\Psi,\lambda} = \sup_{\psi \in \Psi} \text{GCMMD}_{\mu,k_\psi,\lambda}$ yields a good generative model on MNIST. But the linear system of size $M + Md$ is impractical: even on $28 \times 28$ images and using a low-rank approximation, the model took days to converge. We therefore design a less expensive discrepancy in the next section.

The GCMMD is related to some discrepancies previously used in IGM training. The Fisher GAN [34] uses only the variance constraint $\|f\|_{L^2(\mu)}^2 \leq 1$. The Sobolev GAN [33] constrains $\|\nabla f\|_{L^2(\mu)}^2 \leq 1$, along with a vanishing boundary condition on $f$ to ensure a well-defined solution (although this was not used in the implementation, and can cause very unintuitive critic behavior; see Appendix C). The authors considered several choices of $\mu$, including the WGAN-GP measure [18] and mixtures $(\mathbb{P} + \mathbb{Q}_\theta)/2$. Rather than enforcing the constraints in closed form as we do, though, these models used additive regularization. We will compare to the Sobolev GAN in experiments.

## 3.3 Scaled Maximum Mean Discrepancy

We will now derive a lower bound on the Gradient-Constrained MMD which retains many of its attractive qualities but can be estimated in time linear in the dimension $d$.

**Proposition 4.** *Make Assumptions (A) to (D). For any $f \in \mathcal{H}_k$, $\|f\|_{S(\mu),k,\lambda} \leq \sigma_{\mu,k,\lambda}^{-1}\|f\|_{\mathcal{H}_k}$, where*

$$\sigma_{\mu,k,\lambda} := 1 \Big/ \sqrt{\lambda + \int k(x,x)\mu(\mathrm{d}x) + \sum_{i=1}^d \int \frac{\partial^2 k(y,z)}{\partial y_i \partial z_i}\Big|_{(y,z)=(x,x)} \mu(\mathrm{d}x)}.$$

We then define the Scaled Maximum Mean Discrepancy based on this bound of Proposition 4:

$$\text{SMMD}_{\mu,k,\lambda}(\mathbb{P}, \mathbb{Q}) := \sup_{f \,:\, \sigma_{\mu,k,\lambda}^{-1}\|f\|_{\mathcal{H}} \leq 1} \mathbb{E}_{X \sim \mathbb{P}}\left[f(X)\right] - \mathbb{E}_{Y \sim \mathbb{Q}}\left[f(Y)\right] = \sigma_{\mu,k,\lambda}\,\text{MMD}_k(\mathbb{P}, \mathbb{Q}). \tag{7}$$

Because the constraint in the optimization of (7) is more restrictive than in that of (5), we have that $\mathrm{SMMD}_{\mu,k,\lambda}(\mathbb{P},\mathbb{Q}) \leq \mathrm{GCMMD}_{\mu,k,\lambda}(\mathbb{P},\mathbb{Q})$. The Sobolev norm $\|f\|_{S(\mu),\lambda}$, and a fortiori the gradient norm under $\mu$, is thus also controlled for the SMMD critic. We also show in Appendix F.1 that $\mathrm{SMMD}_{\mu,k,\lambda}$ behaves similarly to $\mathrm{GCMMD}_{\mu,k,\lambda}$ on Gaussians.

If $k_\psi = K \circ \phi_\psi$ and $K(a,b) = g(-\|a-b\|^2)$, then $\sigma^{-2}_{k,\mu,\lambda} = \lambda + g(0) + 2|g'(0)| \, \mathbb{E}_\mu\left[\|\nabla\phi_\psi(X)\|_F^2\right]$. Or if $K$ is linear, $K(a,b) = a^\mathsf{T}b$, then $\sigma^{-2}_{k,\mu,\lambda} = \lambda + \mathbb{E}_\mu\left[\|\phi_\psi(X)\|^2 + \|\nabla\phi_\psi(X)\|_F^2\right]$. Estimating these terms based on samples from $\mu$ is straightforward, giving a natural estimator for the SMMD.

Of course, if $\mu$ and $k$ are fixed, the SMMD is simply a constant times the MMD, and so behaves in essentially the same way as the MMD. But optimizing the SMMD over a kernel family $\Psi$, $\mathcal{D}^{\mu,\Psi,\lambda}_{\mathrm{SMMD}}(\mathbb{P},\mathbb{Q}) := \sup_{\psi\in\Psi} \mathrm{SMMD}_{\mu,k_\psi,\lambda}(\mathbb{P},\mathbb{Q})$, gives a distance very different from $\mathcal{D}^\Psi_{\mathrm{MMD}}$ (3).

Figure 1 (b) shows the vector field for the Optimized SMMD loss in Example 1, using the WGAN-GP measure $\mu = \mathrm{Uniform}(0,\theta)$. The optimization surface is far more amenable: in particular the location $C$, which formerly had an extremely small gradient that made learning effectively impossible, now converges very quickly by first reducing the critic gradient until some signal is available. Figure 1 (d) demonstrates that $\mathcal{D}^{\mu,\Psi,\lambda}_{\mathrm{SMMD}}$, like $\mathcal{D}^{\mu,\Psi,\lambda}_{\mathrm{GCMMD}}$ and $\mathcal{D}^{\mu,\Psi,\lambda}_{\mathrm{LipMMD}}$ but in sharp contrast to $\mathcal{D}^\Psi_{\mathrm{MMD}}$, is continuous with respect to the location $\theta$ and provides a strong gradient towards 0.

We can establish that $\mathcal{D}^{\mu,\Psi,\lambda}_{\mathrm{SMMD}}$ is continuous in the Wasserstein topology under some conditions:

**Theorem 1.** *Let $k_\psi = K \circ \phi_\psi$, with $\phi_\psi : \mathcal{X} \to \mathbb{R}^s$ a fully-connected L-layer network with Leaky-ReLU$_\alpha$ activations whose layers do not increase in width, and $K$ satisfying mild smoothness conditions $Q_K < \infty$ (Assumptions (II) to (V) in Appendix A.2). Let $\Psi^\kappa$ be the set of parameters where each layer's weight matrices have condition number $\mathrm{cond}(W^l) = \|W^l\|/\sigma_{\min}(W^l) \leq \kappa < \infty$. If $\mu$ has a density (Assumption (I)), then*

$$\mathcal{D}^{\mu,\Psi^\kappa,\lambda}_{\mathrm{SMMD}}(\mathbb{P},\mathbb{Q}) \leq \frac{Q_K \kappa^{L/2}}{\sqrt{d_L}\alpha^{L/2}} \, \mathcal{W}(\mathbb{P},\mathbb{Q}).$$

*Thus if $\mathbb{P}_n \xrightarrow{\mathcal{W}} \mathbb{P}$, then $\mathcal{D}^{\mu,\Psi^\kappa,\lambda}_{\mathrm{SMMD}}(\mathbb{P}_n,\mathbb{P}) \to 0$, even if $\mu$ is chosen to depend on $\mathbb{P}$ and $\mathbb{Q}$.*

**Uniform bounds vs bounds in expectation**  Controlling $\|\nabla f_\psi\|^2_{L^2(\mu)} = \mathbb{E}_\mu\|\nabla f_\psi(X)\|^2$ does not necessarily imply a bound on $\|f\|_{\mathrm{Lip}} \geq \sup_{x\in\mathcal{X}}\|\nabla f_\psi(X)\|$, and so does not in general give continuity via Proposition 2. Theorem 1 implies that when the network's weights are well-conditioned, it is sufficient to only control $\|\nabla f_\psi\|^2_{L^2(\mu)}$, which is far easier in practice than controlling $\|f\|_{\mathrm{Lip}}$.

If we instead tried to directly controlled $\|f\|_{\mathrm{Lip}}$ with e.g. spectral normalization (SN) [32], we could significantly reduce the expressiveness of the parametric family. In Example 1, constraining $\|\phi_\psi\|_{\mathrm{Lip}} = 1$ limits us to only $\Psi = \{1\}$. Thus $\mathcal{D}^{\{1\}}_{\mathrm{MMD}}$ is simply the MMD with an RBF kernel of bandwidth 1, which has poor gradients when $\theta$ is far from 0 (Figure 1 (c), blue). The Cauchy-Schwartz bound of Proposition 4 allows jointly adjusting the smoothness of $k_\psi$ and the critic $f$, while SN must control the two independently. Relatedly, limiting $\|\phi\|_{\mathrm{Lip}}$ by limiting the Lipschitz norm of each layer could substantially reduce capacity, while $\|\nabla f_\psi\|_{L^2(\mu)}$ need not be decomposed by layer. Another advantage is that $\mu$ provides a data-dependent measure of complexity as in [10]: we do not needlessly prevent ourselves from using critics that behave poorly only far from the data.

**Spectral parametrization**  When the generator is near a local optimum, the critic might identify only one direction on which $\mathbb{Q}_\theta$ and $\mathbb{P}$ differ. If the generator parameterization is such that there is no local way for the generator to correct it, the critic may begin to single-mindedly focus on this difference, choosing redundant convolutional filters and causing the condition number of the weights to diverge. If this occurs, the generator will be motivated to fix this single direction while ignoring all other aspects of the distributions, after which it may become stuck. We can help avoid this collapse by using a critic parameterization that encourages diverse filters with higher-rank weight matrices. Miyato et al. [32] propose to parameterize the weight matrices as $W = \gamma\bar{W}/\|\bar{W}\|_{\mathrm{op}}$, where $\|\bar{W}\|_{\mathrm{op}}$ is the spectral norm of $\bar{W}$. This parametrization works particularly well with $\mathcal{D}^{\mu,\Psi,\lambda}_{\mathrm{SMMD}}$; Figure 2 (b) shows the singular values of the second layer of a critic's network (and Figure 9, in Appendix F.3, shows more layers), while Figure 2 (d) shows the evolution of the condition number during training. The conditioning of the weight matrix remains stable throughout training with spectral parametrization, while it worsens through training in the default case.

# 4 Experiments

We evaluated unsupervised image generation on three datasets: CIFAR-10 [26] (60 000 images, $32 \times 32$), CelebA [29] (202 599 face images, resized and cropped to $160 \times 160$ as in [7]), and the more challenging ILSVRC2012 (ImageNet) dataset [41] (1 281 167 images, resized to $64 \times 64$). Code for all of these experiments is available at `github.com/MichaelArbel/Scaled-MMD-GAN`.

**Losses** All models are based on a scalar-output critic network $\phi_\psi : \mathcal{X} \to \mathbb{R}$, except MMDGAN-GP where $\phi_\psi : \mathcal{X} \to \mathbb{R}^{16}$ as in [7]. The WGAN and Sobolev GAN use a critic $f = \phi_\psi$, while the GAN uses a discriminator $D_\psi(x) = 1/(1 + \exp(-\phi_\psi(x)))$. The MMD-based methods use a kernel $k_\psi(x, y) = \exp(-(\phi_\psi(x) - \phi_\psi(y))^2/2)$, except for MMDGAN-GP which uses a mixture of RQ kernels as in [7]. Increasing the output dimension of the critic or using a different kernel didn't substantially change the performance of our proposed method. We also consider SMMD with a linear top-level kernel, $k(x, y) = \phi_\psi(x)\phi_\psi(y)$; because this becomes essentially identical to a WGAN (Appendix E), we refer to this method as SWGAN. SMMD and SWGAN use $\mu = \mathbb{P}$; Sobolev GAN uses $\mu = (\mathbb{P} + \mathbb{Q})/2$ as in [33]. We choose $\lambda$ and an overall scaling to obtain the losses:

$$\text{SMMD:} \frac{\widehat{\text{MMD}}^2_{k_\psi}(\mathbb{P}, \mathbb{Q}_\theta)}{1 + 10\,\mathbb{E}_{\hat{\mathbb{P}}}\left[\|\nabla\phi_\psi(X)\|^2_F\right]}, \quad \text{SWGAN:} \frac{\mathbb{E}_{\hat{\mathbb{P}}}\left[\phi_\psi(X)\right] - \mathbb{E}_{\hat{\mathbb{Q}}_\theta}\left[\phi_\psi(X)\right]}{\sqrt{1 + 10\left(\mathbb{E}_{\hat{\mathbb{P}}}\left[|\phi_\psi(X)|^2\right] + \mathbb{E}_{\hat{\mathbb{P}}}\left[\|\nabla\phi_\psi(X)\|^2_F\right]\right)}}.$$

**Architecture** For CIFAR-10, we used the CNN architecture proposed by [32] with a 7-layer critic and a 4-layer generator. For CelebA, we used a 5-layer DCGAN discriminator and a 10-layer ResNet generator as in [7]. For ImageNet, we used a 10-layer ResNet for both the generator and discriminator. In all experiments we used 64 filters for the smallest convolutional layer, and double it at each layer (CelebA/ImageNet) or every other layer (CIFAR-10). The input codes for the generator are drawn from Uniform $([-1, 1]^{128})$. We consider two parameterizations for each critic: a standard one where the parameters can take any real value, and a spectral parametrization (denoted SN-) as above [32]. Models without explicit gradient control (SN-GAN, SN-MMDGAN, SN-MMGAN-L2, SN-WGAN) fix $\gamma = 1$, for spectral normalization; others learn $\gamma$, using a spectral parameterization.

**Training** All models were trained for 150 000 generator updates on a single GPU, except for ImageNet where the model was trained on 3 GPUs simultaneously. To limit communication overhead we averaged the MMD estimate on each GPU, giving the block MMD estimator [54]. We always used 64 samples per GPU from each of $\mathbb{P}$ and $\mathbb{Q}$, and 5 critic updates per generator step. We used initial learning rates of 0.0001 for CIFAR-10 and CelebA, 0.0002 for ImageNet, and decayed these rates using the KID adaptive scheme of [7]: every 2 000 steps, generator samples are compared to those from 20 000 steps ago, and if the relative KID test [9] fails to show an improvement three consecutive times, the learning rate is decayed by 0.8. We used the Adam optimizer [25] with $\beta_1 = 0.5$, $\beta_2 = 0.9$.

**Evaluation** To compare the sample quality of different models, we considered three different scores based on the Inception network [49] trained for ImageNet classification, all using default parameters in the implementation of [7]. The *Inception Score (IS)* [42] is based on the entropy of predicted labels; higher values are better. Though standard, this metric has many issues, particularly on datasets other than ImageNet [4, 7, 20]. The *FID* [20] instead measures the similarity of samples from the generator and the target as the Wasserstein-2 distance between Gaussians fit to their intermediate representations. It is more sensible than the IS and becoming standard, but its estimator is strongly biased [7]. The *KID* [7] is similar to FID, but by using a polynomial-kernel MMD its estimates enjoy better statistical properties and are easier to compare. (A similar score was recommended by [21].)

**Results** Table 1a presents the scores for models trained on both CIFAR-10 and CelebA datasets. On CIFAR-10, SN-SWGAN and SN-SMMDGAN performed comparably to SN-GAN. But on CelebA, SN-SWGAN and SN-SMMDGAN dramatically outperformed the other methods with the same architecture in all three metrics. It also trained faster, and consistently outperformed other methods over multiple initializations (Figure 2 (a)). It is worth noting that SN-SWGAN far outperformed WGAN-GP on both datasets. Table 1b presents the scores for SMMDGAN and SN-SMMDGAN trained on ImageNet, and the scores of pre-trained models using BGAN [6] and SN-GAN [32].[6] The

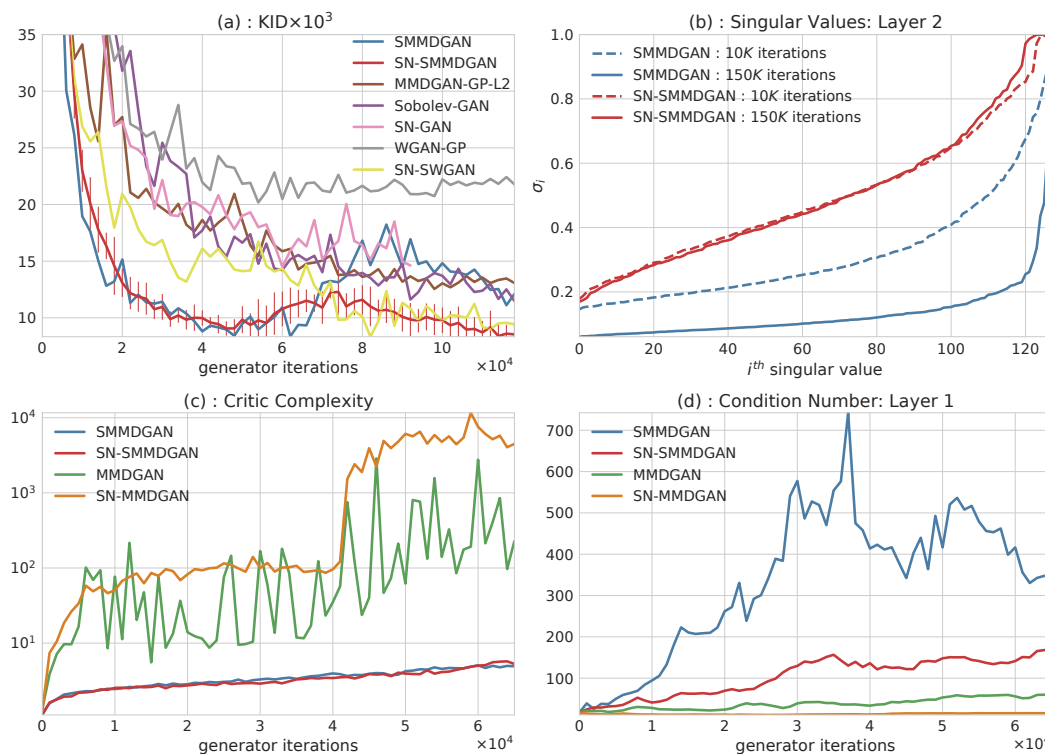

Figure 2: The training process on CelebA. (a) KID scores. We report a final score for SN-GAN slightly before its sudden failure mode; MMDGAN and SN-MMDGAN were unstable and had scores around 100. (b) Singular values of the second layer, both early (dashed) and late (solid) in training. (c) $\sigma_{\mu,k,\lambda}^{-2}$ for several MMD-based methods. (d) The condition number in the first layer through training. SN alone does not control $\sigma_{\mu,k,\lambda}$, and SMMD alone does not control the condition number.

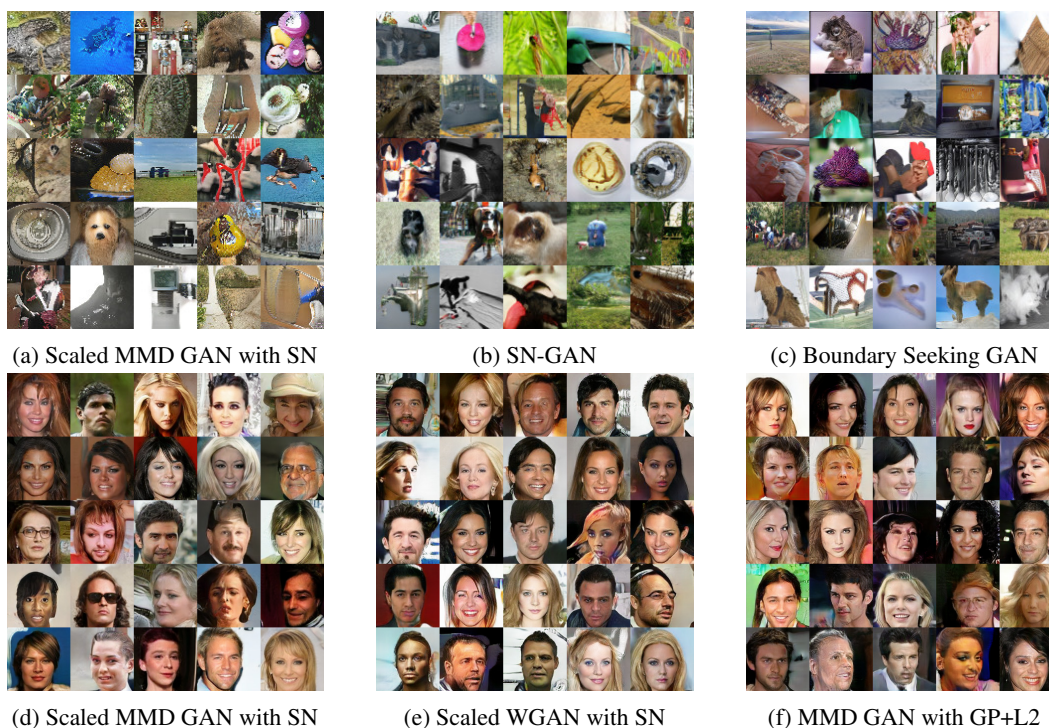

(a) Scaled MMD GAN with SN     (b) SN-GAN     (c) Boundary Seeking GAN

(d) Scaled MMD GAN with SN     (e) Scaled WGAN with SN     (f) MMD GAN with GP+L2

Figure 3: Samples from various models. Top: $64 \times 64$ ImageNet; bottom: $160 \times 160$ CelebA.

Table 1: Mean (standard deviation) of score estimates, based on $50\,000$ samples from each model.

(a) CIFAR-10 and CelebA.

| Method | CIFAR-10 | | | CelebA | | |
|---|---|---|---|---|---|---|
| | IS | FID | KID$\times 10^3$ | IS | FID | KID$\times 10^3$ |
| WGAN-GP | 6.9±0.2 | 31.1±0.2 | 22.2±1.1 | 2.7±0.0 | 29.2±0.2 | 22.0±1.0 |
| MMDGAN-GP-L2 | 6.9±0.1 | 31.4±0.3 | 23.3±1.1 | 2.6±0.0 | 20.5±0.2 | 13.0±1.0 |
| Sobolev-GAN | 7.0±0.1 | 30.3±0.3 | 22.3±1.2 | **2.9±0.0** | 16.4±0.1 | 10.6±0.5 |
| SMMDGAN | 7.0±0.1 | 31.5±0.4 | 22.2±1.1 | 2.7±0.0 | 18.4±0.2 | 11.5±0.8 |
| SN-GAN | **7.2±0.1** | 26.7±0.2 | **16.1±0.9** | 2.7±0.0 | 22.6±0.1 | 14.6±1.1 |
| SN-SWGAN | **7.2±0.1** | 28.5±0.2 | **17.6±1.1** | 2.8±0.0 | 14.1±0.2 | 7.7±0.5 |
| SN-SMMDGAN | **7.3±0.1** | **25.0±0.3** | **16.6±2.0** | 2.8±0.0 | **12.4±0.2** | **6.1±0.4** |

(b) ImageNet.

| Method | IS | FID | KID$\times 10^3$ |
|---|---|---|---|
| BGAN | 10.7±0.4 | 43.9±0.3 | 47.0±1.1 |
| SN-GAN | **11.2±0.1** | 47.5±0.1 | 44.4±2.2 |
| SMMDGAN | 10.7±0.2 | 38.4±0.3 | 39.3±2.5 |
| SN-SMMDGAN | 10.9±0.1 | **36.6±0.2** | **34.6±1.6** |

proposed methods substantially outperformed both methods in FID and KID scores. Figure 3 shows samples on ImageNet and CelebA; Appendix F.4 has more.

**Spectrally normalized WGANs / MMDGANs** To control for the contribution of the spectral parametrization to the performance, we evaluated variants of MMDGANs, WGANs and Sobolev-GAN using spectral normalization (in Table 2, Appendix F.3). WGAN and Sobolev-GAN led to unstable training and didn't converge at all (Figure 11) despite many attempts. MMDGAN converged on CIFAR-10 (Figure 11) but was unstable on CelebA (Figure 10). The gradient control due to SN is thus probably too loose for these methods. This is reinforced by Figure 2 (c), which shows that the expected gradient of the critic network is much better-controlled by SMMD, even when SN is used. We also considered variants of these models with a learned $\gamma$ while also adding a gradient penalty and an $L_2$ penalty on critic activations [7, footnote 19]. These generally behaved similarly to MMDGAN, and didn't lead to substantial improvements. We ran the same experiments on CelebA, but aborted the runs early when it became clear that training was not successful.

**Rank collapse** We occasionally observed the failure mode for SMMD where the critic becomes low-rank, discussed in Section 3.3, especially on CelebA; this failure was obvious even in the training objective. Figure 2 (b) is one of these examples. Spectral parametrization seemed to prevent this behavior. We also found one could avoid collapse by reverting to an earlier checkpoint and increasing the RKHS regularization parameter $\lambda$, but did not do this for any of the experiments here.

## 5 Conclusion

We studied gradient regularization for MMD-based critics in implicit generative models, clarifying how previous techniques relate to the $\mathcal{D}^{\Psi}_{\mathrm{MMD}}$ loss. Based on these insights, we proposed the Gradient-Constrained MMD and its approximation the Scaled MMD, a new loss function for IGMs that controls gradient behavior in a principled way and obtains excellent performance in practice.

One interesting area of future study for these distances is their behavior when used to diffuse particles distributed as $\mathbb{Q}$ towards particles distributed as $\mathbb{P}$. Mroueh et al. [33, Appendix A.1] began such a study for the Sobolev GAN loss; [35] proved convergence and studied discrete-time approximations.

Another area to explore is the geometry of these losses, as studied by Bottou et al. [8], who showed potential advantages of the Wasserstein geometry over the MMD. Their results, though, do not address any distances based on optimized kernels; the new distances introduced here might have interesting geometry of their own.

## Footnotes

[2] $f\#\mathbb{P}$ denotes the *pushforward* of a distribution: if $X \sim \mathbb{P}$, then $f(X) \sim f\#\mathbb{P}$.

[3] [27, Theorem 4] makes a similar claim to Proposition 2, but its proof was incorrect: it tries to uniformly bound $\mathrm{MMD}_{k_\psi} \leq \mathcal{W}^2$, but the bound used is for a Wasserstein in terms of $\|k_\psi(x, \cdot) - k_\psi(y, \cdot)\|_{\mathcal{H}_{k_\psi}}$.

[4]We use $(m, i)$ to denote $(m - 1)d + i$; thus $\nabla\eta(X)$ stacks $\nabla\eta(X_1), \ldots, \nabla\eta(X_M)$ into one vector.

[5]We use $\partial_i k(x, y)$ to denote the partial derivative with respect to $x_i$, and $\partial_{i+d} k(x, y)$ that for $y_i$.

[6]These models are courtesy of the respective authors and also trained at $64 \times 64$ resolution. SN-GAN used the same architecture as our model, but trained for 250 000 generator iterations; BS-GAN used a similar 5-layer ResNet architecture and trained for 74 epochs, comparable to SN-GAN.

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
