[Supplementary Material]

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

# A  Proofs

We first review some basic properties of Reproducing Kernel Hilbert Spaces. We consider here a separable RKHS $\mathcal{H}$ with basis $(e_i)_{i \in I}$, where $I$ is either finite if $\mathcal{H}$ is finite-dimensional, or $I = \mathbb{N}$ otherwise. We also assume that the reproducing kernel $k$ is continuously twice differentiable.

We use a slightly nonstandard notation for derivatives: $\partial_i f(x)$ denotes the $i$th partial derivative of $f$ evaluated at $x$, and $\partial_i \partial_{j+d} k(x, y)$ denotes $\frac{\partial^2 k(a,b)}{\partial a_i \partial b_j}|_{(a,b)=(x,y)}$.

Then the following reproducing properties hold for any given function $f$ in $\mathcal{H}$ [47, Lemma 4.34]:

$$f(x) = \langle f, k(x, .) \rangle_{\mathcal{H}} \tag{8}$$
$$\partial_i f(x) = \langle f, \partial_i k(x, .) \rangle_{\mathcal{H}}. \tag{9}$$

We say that an operator $A : \mathcal{H} \mapsto \mathcal{H}$ is Hilbert-Schmidt if $\|A\|_{HS}^2 = \sum_{i \in I} \|Ae_i\|_{\mathcal{H}}^2$ is finite. $\|A\|_{HS}$ is called the Hilbert-Schmidt norm of $A$. The space of Hilbert-Schmidt operators itself a Hilbert space with the inner product $\langle A, B \rangle_{HS} = \sum_{i \in I} \langle Ae_i, Be_i \rangle_{\mathcal{H}}$. Moreover, we say that an operator $A$ is trace-class if its trace norm is finite, i.e. $\|A\|_1 = \sum_{i \in I} \langle e_i, (A^*A)^{\frac{1}{2}} e_i \rangle_{\mathcal{H}} < \infty$. The outer product $f \otimes g$ for $f, g \in \mathcal{H}$ gives an $\mathcal{H} \to \mathcal{H}$ operator such that $(f \otimes g)v = \langle g, v \rangle_{\mathcal{H}} f$ for all $v$ in $\mathcal{H}$.

Given two vectors $f$ and $g$ in $\mathcal{H}$ and a Hilbert-Schmidt operator $A$ we have the following properties:

(i) The outer product $f \otimes g$ is a Hilbert-Schmidt operator with Hilbert-Schmidt norm given by: $\|f \otimes g\|_{HS} = \|f\|_{\mathcal{H}} \|g\|_{\mathcal{H}}$.

(ii) The inner product between two rank-one operators $f \otimes g$ and $u \otimes v$ is $\langle f \otimes g, u \otimes v \rangle_{HS} = \langle f, u \rangle_{\mathcal{H}} \langle g, v \rangle_{\mathcal{H}}$.

(iii) The following identity holds: $\langle f, Ag \rangle_{\mathcal{H}} = \langle f \otimes g, A \rangle_{HS}$.

Define the following covariance-type operators:

$$D_x = k(x, \cdot) \otimes k(x, \cdot) + \sum_{i=1}^{d} \partial_i k(x, \cdot) \otimes \partial_i k(x, \cdot) \quad D_\mu = \mathbb{E}_{X \sim \mu} D_X \quad D_{\mu,\lambda} = D_\mu + \lambda I; \tag{10}$$

these are useful in that, using (8) and (9), $\langle f, D_x g \rangle = f(x)g(x) + \sum_{i=1}^{d} \partial_i f(x) \, \partial_i g(x)$.

## A.1  Definitions and estimators of the new distances

We will need the following assumptions about the distributions $\mathbb{P}$ and $\mathbb{Q}$, the measure $\mu$, and the kernel $k$:

(A) $\mathbb{P}$ and $\mathbb{Q}$ have integrable first moments.

(B) $\sqrt{k(x,x)}$ grows at most linearly in $x$: for all $x$ in $\mathcal{X}$, $\sqrt{k(x,x)} \leq C(\|x\| + 1)$ for some constant $C$.

(C) The kernel $k$ is twice continuously differentiable.

(D) The functions $x \mapsto k(x, x)$ and $x \mapsto \partial_i \partial_{i+d} k(x, x)$ for $1 \leq i \leq d$ are $\mu$-integrable.

When $k = K \circ \phi_\psi$, Assumption (B) is automatically satisfied by a $K$ such as the Gaussian; when $K$ is linear, it is true for a quite general class of networks $\phi_\psi$ [7, Lemma 1].

We will first give a form for the Gradient-Constrained MMD (5) in terms of the operator (10):

**Proposition 5.** *Under Assumptions (A) to (D), the Gradient-Constrained MMD is given by*

$$\text{GCMMD}_{\mu,k,\lambda}(\mathbb{P}, \mathbb{Q}) = \sqrt{\langle \eta, D_{\mu,\lambda}^{-1} \eta \rangle_{\mathcal{H}}}. \tag{11}$$

*Proof of Proposition 5.* Let $f$ be a function in $\mathcal{H}$. We will first express the squared $\lambda$-regularized Sobolev norm of $f$ (6) as a quadratic form in $\mathcal{H}$. Recalling the reproducing properties of (8) and (9), we have:

$$\|f\|_{S(\mu),k,\lambda}^2 = \int \langle f, k(x, \cdot) \rangle_{\mathcal{H}}^2 \, \mu(\mathrm{d}x) + \sum_{i=1}^{d} \int \langle f, \partial_i k(x, \cdot) \rangle_{\mathcal{H}}^2 \, \mu(\mathrm{d}x) + \lambda \|f\|_{\mathcal{H}}^2.$$

Using Property (ii) and the operator (10), one further gets

$$\|f\|^2_{S(\mu),k,\lambda} = \int \langle f \otimes f, D_x \rangle_{\text{HS}} \, \mu(\mathrm{d}x) + \lambda \|f\|^2_{\mathcal{H}}.$$

Under Assumption (D), and using Lemma 6, one can take the integral inside the inner product, which leads to $\|f\|^2_{S(\mu),k,\lambda} = \langle f \otimes f, D_\mu \rangle_{\text{HS}} + \lambda \|f\|^2_{\mathcal{H}}$. Finally, using Property (iii) it follows that

$$\|f\|^2_{S(\mu),k,\lambda} = \langle f, D_{\mu,\lambda} f \rangle_{\mathcal{H}}.$$

Under Assumptions (A) and (B), Lemma 6 applies, and it follows that $k(x, \cdot)$ is also Bochner integrable under $\mathbb{P}$ and $\mathbb{Q}$. Thus

$$\mathbb{E}_{\mathbb{P}} \left[ \langle f, k(x, \cdot) \rangle_{\mathcal{H}} \right] - \mathbb{E}_{\mathbb{Q}} \left[ \langle f, k(x, \cdot) \rangle_{\mathcal{H}} \right] = \langle f, \mathbb{E}_{\mathbb{P}}\left[ k(x, \cdot) \right] - \mathbb{E}_{\mathbb{P}}\left[ k(x, \cdot) \right] \rangle_{\mathcal{H}} = \langle f, \eta \rangle_{\mathcal{H}},$$

where $\eta$ is defined as this difference in mean embeddings.

Since $D_{\mu,\lambda}$ is symmetric positive definite, its square-root $D^{\frac{1}{2}}_{\mu,\lambda}$ is well-defined and is also invertible. For any $f \in \mathcal{H}$, let $g = D^{\frac{1}{2}}_{\mu,\lambda} f$, so that $\langle f, D_{\mu,\lambda} f \rangle_{\mathcal{H}} = \|g\|^2_{\mathcal{H}}$. Note that for any $g \in \mathcal{H}$, there is a corresponding $f = D^{-\frac{1}{2}}_{\mu,\lambda} g$. Thus we can re-express the maximization problem in (5) in terms of $g$:

$$\begin{aligned} \text{GCMMD}_{\mu,k,\lambda}(\mathbb{P}, \mathbb{Q}) &:= \sup_{\substack{f \in \mathcal{H} \\ \langle f, D_{\mu,\lambda} f \rangle_{\mathcal{H}} \leq 1}} \langle f, \eta \rangle_{\mathcal{H}} = \sup_{\substack{g \in \mathcal{H} \\ \|g\|_{\mathcal{H}} \leq 1}} \langle D^{-\frac{1}{2}}_{\mu,\lambda} g, \eta \rangle_{\mathcal{H}} \\ &= \sup_{\substack{g \in \mathcal{H} \\ \|g\|_{\mathcal{H}} \leq 1}} \langle g, D^{-\frac{1}{2}}_{\mu,\lambda} \eta \rangle_{\mathcal{H}} = \|D^{-\frac{1}{2}}_{\mu,\lambda} \eta\|_{\mathcal{H}} = \sqrt{\langle \eta, D^{-1}_{\mu,\lambda} \eta \rangle_{\mathcal{H}}}. \qquad \square \end{aligned}$$

Proposition 5, though, involves inverting the infinite-dimensional operator $D_{\mu,\lambda}$ and thus doesn't directly give us a computable estimator. Proposition 3 solves this problem in the case where $\mu$ is a discrete measure:

**Proposition 3.** *Let $\hat{\mu} = \sum_{m=1}^{M} \delta_{X_m}$ be an empirical measure of $M$ points. Let $\eta(X) \in \mathbb{R}^M$ have $m$th entry $\eta(X_m)$, and $\nabla \eta(X) \in \mathbb{R}^{Md}$ have $(m, i)$th entry[7] $\partial_i \eta(X_m)$. Then under Assumptions (A) to (D), the Gradient-Constrained MMD is*

$$\text{GCMMD}^2_{\hat{\mu},k,\lambda}(\mathbb{P}, \mathbb{Q}) = \frac{1}{\lambda} \left( \text{MMD}^2(\mathbb{P}, \mathbb{Q}) - \bar{P}(\eta) \right)$$

$$\bar{P}(\eta) = \begin{bmatrix} \eta(X) \\ \nabla \eta(X) \end{bmatrix}^{\mathsf{T}} \left( \begin{bmatrix} K & G^{\mathsf{T}} \\ G & H \end{bmatrix} + M\lambda I_{M+Md} \right)^{-1} \begin{bmatrix} \eta(X) \\ \nabla \eta(X) \end{bmatrix},$$

*where $K$ is the kernel matrix $K_{m,m'} = k(X_m, X_{m'})$, $G$ is the matrix of left derivatives $G_{(m,i),m'} = \partial_i k(X_m, X_{m'})$, and $H$ that of derivatives of both arguments $H_{(m,i),(m',j)} = \partial_i \partial_{j+d} k(X_m, X_{m'})$.*

Before proving Proposition 3, we note the following interesting alternate form. Let $\bar{e}_i$ be the $i$th standard basis vector for $\mathbb{R}^{M+Md}$, and define $T : \mathcal{H} \to \mathbb{R}^{M+Md}$ as the linear operator

$$T = \sum_{m=1}^{M} \bar{e}_m \otimes k(X_m, \cdot) + \sum_{m=1}^{M} \sum_{i=1}^{d} \bar{e}_{m+(m,i)} \otimes \partial_i k(X_m, \cdot).$$

Then $\begin{bmatrix} \eta(X) \\ \nabla \eta(X) \end{bmatrix} = T\eta$, and $\begin{bmatrix} K & G^{\mathsf{T}} \\ G & H \end{bmatrix} = TT^*$. Thus we can write

$$\text{GCMMD}^2_{\hat{\mu},k,\lambda} = \frac{1}{\lambda} \left\langle \eta, \left( I - T^*(TT^* + M\lambda I)^{-1} T \right) \eta \right\rangle_{\mathcal{H}}.$$

*Proof of Proposition 3.* Let $g \in \mathcal{H}$ be the solution to the regression problem $D_{\mu,\lambda} g = \eta$:

$$\frac{1}{M} \sum_{m=1}^{M} \left[ g(X_m) k(X_m, \cdot) + \sum_{i=1}^{d} \partial_i g(X_m) \partial_i k(X_m, \cdot) \right] + \lambda g = \eta$$

$$g = \frac{1}{\lambda} \eta - \frac{1}{\lambda M} \sum_{m=1}^{M} \left[ g(X_m) k(X_m, \cdot) + \sum_{i=1}^{d} \partial_i g(X_m) \partial_i k(X_m, \cdot) \right]. \tag{12}$$

Taking the inner product of both sides of (12) with $k(X_{m'}, \cdot)$ for each $1 \leq m' \leq M$ yields the following $M$ equations:

$$g(X_{m'}) = \frac{1}{\lambda} \eta(X_{m'}) - \frac{1}{\lambda M} \sum_{m=1}^{M} \left[ g(X_m) K_{m,m'} + \sum_{i=1}^{d} \partial_i g(X_m) G_{(m,i),m'} \right]. \tag{13}$$

Doing the same with $\partial_j k(X_{m'}, \cdot)$ gives $Md$ equations:

$$\partial_j g(X_{m'}) = \frac{1}{\lambda} \partial_j \eta(X_{m'}) - \frac{1}{\lambda M} \sum_{m=1}^{M} \left[ g(X_m) G_{(m',j),m} + \sum_{i=1}^{d} \partial_i g(X_m) H_{(m,i),(m',j)} \right]. \tag{14}$$

From (12), it is clear that $g$ is a linear combination of the form:

$$g(x) = \frac{1}{\lambda} \eta(x) - \frac{1}{\lambda M} \sum_{m=1}^{M} \left[ \alpha_m k(X_m, x) + \sum_{i=1}^{d} \beta_{m,i} \partial_i k(X_m, x) \right],$$

where the coefficients $\alpha := (\alpha_m = g(X_m))_{1 \leq m \leq M}$ and $\beta := (\beta_{m,i} = \partial_i g(X_m))_{\substack{1 \leq m \leq M \\ 1 \leq i \leq d}}$ satisfy the system of equations (13) and (14). We can rewrite this system as

$$\begin{bmatrix} K + M\lambda I_M & G^\mathsf{T} \\ G & H + M\lambda I_{Md} \end{bmatrix} \begin{bmatrix} \alpha \\ \beta \end{bmatrix} = M \begin{bmatrix} \eta(X) \\ \nabla \eta(X) \end{bmatrix},$$

where $I_M$, $I_{Md}$ are the identity matrices of dimension $M$, $Md$. Since $K$ and $H$ must be positive semidefinite, an inverse exists. We conclude by noticing that

$$\mathrm{GCMMD}_{\hat{\mu},k,\lambda}(\mathbb{P}, \mathbb{Q})^2 = \langle \eta, g \rangle_{\mathcal{H}} = \frac{1}{\lambda} \|\eta\|_{\mathcal{H}}^2 - \frac{1}{\lambda M} \sum_{m=1}^{M} \left[ \alpha_m \eta(X_m) + \sum_{i=1}^{d} \beta_{m,i} \partial_i \eta(X_m) \right]. \quad \square$$

The following result was key to our definition of the SMMD in Section 3.3.

**Proposition 4.** *Under Assumptions (A) to (D), we have for all $f \in \mathcal{H}$ that*

$$\|f\|_{S(\mu),k,\lambda} \leq \sigma_{\mu,k,\lambda}^{-1} \|f\|_{\mathcal{H}_k},$$

*where* $\sigma_{k,\mu,\lambda} := 1/\sqrt{\lambda + \int k(x,x) \mu(\mathrm{d}x) + \sum_{i=1}^{d} \int \partial_i \partial_{i+d} k(x,x) \mu(\mathrm{d}x)}$.

*Proof of Proposition 4.* The key idea here is to use the Cauchy-Schwarz inequality for the Hilbert-Schmidt inner product. Letting $f \in \mathcal{H}$, $\|f\|_{S(\mu),k,\lambda}^2$ is

$$\int f(x)^2 \, \mu(\mathrm{d}x) + \int \|\nabla f(x)\|^2 \, \mu(\mathrm{d}x) + \lambda \|f\|_{\mathcal{H}}^2$$

$$\overset{(a)}{=} \int \langle f, k(x, \cdot) \otimes k(x, \cdot) f \rangle_{\mathcal{H}} \, \mu(\mathrm{d}x) + \sum_{i=1}^{d} \int \langle f, \partial_i k(x, \cdot) \otimes \partial_i k(x, \cdot) f \rangle_{\mathcal{H}} \, \mu(\mathrm{d}x) + \lambda \|f\|_{\mathcal{H}}^2$$

$$\overset{(b)}{=} \int \langle f \otimes f, k(x, \cdot) \otimes k(x, \cdot) \rangle_{\mathrm{HS}} \, \mu(\mathrm{d}x) + \sum_{i=1}^{d} \int \langle f \otimes f, \partial_i k(x, \cdot) \otimes \partial_i k(x, \cdot) \rangle_{\mathrm{HS}} \, \mu(\mathrm{d}x) + \lambda \|f\|_{\mathcal{H}}^2$$

$$\overset{(c)}{\leq} \|f\|_{\mathcal{H}}^2 \left[ \int k(x,x) \, \mu(\mathrm{d}x) + \sum_{i=1}^{d} \int \partial_i \partial_{i+d} k(x,x) \, \mu(\mathrm{d}x) + \lambda \right].$$

$(a)$ follows from the reproducing properties (8) and (9) and Property (ii). $(b)$ is obtained using Property (iii), while $(c)$ follows from the Cauchy-Schwarz inequality and Property (i). $\square$

**Lemma 6.** *Under Assumption ($\mathbf{D}$), $D_x$ is Bochner integrable and its integral $D_\mu$ is a trace-class symmetric positive semi-definite operator with $D_{\mu,\lambda} = D + \lambda I$ invertible for any positive $\lambda$. Moreover, for any Hilbert-Schmidt operator $A$ we have: $\langle A, D_\mu \rangle_{HS} = \int \langle A, D_x \rangle_{HS} \mu(\mathrm{d}x)$.*

*Under Assumptions ($\mathbf{A}$) and ($\mathbf{B}$), $k(x,\cdot)$ is Bochner integrable with respect to any probability distribution $\mathbb{P}$ with finite first moment and the following relation holds: $\langle f, \mathbb{E}_{\mathbb{P}}[k(x,\cdot)] \rangle_{\mathcal{H}} = \mathbb{E}_{\mathbb{P}}[\langle f, k(x,\cdot) \rangle_{\mathcal{H}}]$ for all $f$ in $\mathcal{H}$.*

*Proof.* The operator $D_x$ is positive self-adjoint. It is also trace-class, as by the triangle inequality

$$\|D_x\|_1 \le \|k(x,\cdot) \otimes k(x,\cdot)\|_1 + \sum_{i=1}^d \|\partial_i k(x,\cdot) \otimes \partial_i k(x,\cdot)\|_1$$

$$= \|k(x,\cdot)\|_{\mathcal{H}}^2 + \sum_{i=1}^d \|\partial_i k(x,\cdot)\|_{\mathcal{H}}^2 < \infty.$$

By Assumption ($\mathbf{D}$), we have that $\int \|D_x\|_1 \, \mu(\mathrm{d}x) < \infty$ which implies that $D_x$ is $\mu$-integrable in the Bochner sense [39, Definition 1 and Theorem 2]. Its integral $D_\mu$ is trace-class and satisfies $\|D_\mu\|_1 \le \int \|D_x\|_1 \, \mu(\mathrm{d}x)$. This allows to have $\langle A, D_\mu \rangle_{HS} = \int \langle A, D_x \rangle_{HS} \, \mu(\mathrm{d}x)$ for all Hilbert-Schmidt operators $A$. Moreover, the integral preserves the symmetry and positivity. It follows that $\mathcal{D}_{\mu,\lambda}$ is invertible.

The Bochner integrability of $k(x,\cdot)$ under a distribution $\mathbb{P}$ with finite moment follows directly from Assumptions ($\mathbf{A}$) and ($\mathbf{B}$), since $\int \|k(x,\cdot)\| \, \mathbb{P}(\mathrm{d}x) \le C \int (\|x\| + 1) \, \mathbb{P}(\mathrm{d}x) < \infty$. This allows us to write $\langle f, \mathbb{E}_{\mathbb{P}}[k(x,\cdot)] \rangle_{\mathcal{H}} = \mathbb{E}_{\mathbb{P}}[\langle f, k(x,\cdot) \rangle_{\mathcal{H}}]$. $\qquad \square$

### A.2 Continuity of the Optimized Scaled MMD in the Wasserstein topology

To prove Theorem 1, we we will first need some new notation.

We assume the kernel is $k = K \circ \phi_\psi$, i.e. $k_\psi(x,y) = K(\phi_\psi(x), \phi_\psi(y))$, where the representation function $\phi_\psi$ is a network $\phi_\psi(X) : \mathbb{R}^d \to \mathbb{R}^{d_L}$ consisting of $L$ fully-connected layers:

$$
\begin{aligned}
h_\psi^0(X) &= X \\
h_\psi^l(X) &= W^l \sigma_{l-1}(h_\psi^{l-1}(X)) + b^l \qquad \text{for } 1 \le l \le L \\
\phi_\psi(X) &= h_\psi^L(X).
\end{aligned}
\tag{15}
$$

The intermediate representations $h_\psi^l(X)$ are of dimension $d_l$, the weights $W^l$ are matrices in $\mathbb{R}^{d_l \times d_{l-1}}$, and biases $b^l$ are vectors in $\mathbb{R}^{d_l}$. The elementwise activation function $\sigma$ is given by $\sigma_0(x) = x$, and for $l > 0$ the activation $\sigma_l$ is a leaky ReLU with leak coefficient $0 < \alpha < 1$:

$$\sigma_l(x) = \sigma(x) = \begin{cases} x & x > 0 \\ \alpha x & x \le 0 \end{cases} \qquad \text{for } l > 0. \tag{16}$$

The parameter $\psi$ is the concatenation of all the layer parameters:

$$\psi = \left( (W^L, b^L), (W^{L-1}, b^{L-1}), \dots, (W^1, b^1) \right).$$

We denote by $\Psi$ the set of all such possible parameters, i.e. $\Psi = \mathbb{R}^{d_L \times d_{L-1}} \times \mathbb{R}^{d_L} \times \cdots \times \mathbb{R}^{d_1 \times d} \times \mathbb{R}^{d_1}$. Define the following restrictions of $\Psi$:

$$\Psi^\kappa := \left\{ \psi \in \Psi \mid \forall 1 \le l \le L, \ \mathrm{cond}(W^l) \le \kappa \right\} \tag{17}$$

$$\Psi_1^\kappa := \left\{ \psi \in \Psi^\kappa \mid \forall 1 \le l \le L, \ \|W^l\| = 1 \right\}. \tag{18}$$

$\Psi^\kappa$ is the set of those parameters such that $W^l$ have a small condition number, $\mathrm{cond}(W) = \sigma_{\max}(W)/\sigma_{\min}(W)$. $\Psi_1^\kappa$ is the set of per-layer normalized parameters with a condition number bounded by $\kappa$.

Recall the definition of Scaled MMD, (7), where $\lambda > 0$ and $\mu$ is a probability measure:

$$\mathrm{SMMD}_{\mu,k,\lambda}(\mathbb{P},\mathbb{Q}) := \sigma_{\mu,k,\lambda}\,\mathrm{MMD}_k(\mathbb{P},\mathbb{Q})$$

$$\sigma_{k,\mu,\lambda} := 1/\sqrt{\lambda + \int k(x,x)\,\mu(\mathrm{d}x) + \sum_{i=1}^{d} \int \partial_i \partial_{i+d} k(x,x)\,\mu(\mathrm{d}x)}.$$

The Optimized SMMD over the restricted set $\Psi^\kappa$ is given by:

$$\mathcal{D}_{\mathrm{SMMD}}^{\mu,\Psi^\kappa,\lambda}(\mathbb{P},\mathbb{Q}) := \sup_{\psi \in \Psi^\kappa} \mathrm{SMMD}_{\mu,k_\psi,\lambda}\,.$$

The constraint to $\psi \in \Psi^\kappa$ is critical to the proof. In practice, using a spectral parametrization helps enforce this assumption, as shown in Figures 2 and 9. Other regularization methods, like orthogonal normalization [11], are also possible.

We will use the following assumptions:

(I) $\mu$ is a probability distribution absolutely continuous with respect to the Lebesgue measure.

(II) The dimensions of the weights are decreasing per layer: $d_{l+1} \le d_l$ for all $0 \le l \le L-1$.

(III) The non-linearity used is Leaky-ReLU, (16), with leak coefficient $\alpha \in (0,1)$.

(IV) The top-level kernel $K$ is globally Lipschitz in the RKHS norm: there exists a positive constant $L_K > 0$ such that $\|K(a,.) - K(b,.)\| \le L_K\|a-b\|$ for all $a$ and $b$ in $\mathbb{R}^{d_L}$.

(V) There is some $\gamma_K > 0$ for which $K$ satisfies

$$\nabla_b \nabla_c K(b,c)\big|_{(b,c)=(a,a)} \succeq \gamma^2 I \qquad \text{for all } a \in \mathbb{R}^{d_L}. \tag{19}$$

Assumption (I) ensures that the points where $\phi_\psi(X)$ is not differentiable are reached with probability 0 under $\mu$. This assumption can be easily satisfied e.g. if we define $\mu$ by adding Gaussian noise to $\mathbb{P}$.

Assumption (II) helps ensure that the span of $W^l$ is never contained in the null space of $W^{l+1}$. Using Leaky-ReLU as a non-linearity, Assumption (III), further ensures that the network $\phi_\psi$ is locally full-rank almost everywhere; this might not be true with ReLU activations, where it could be always 0. Assumptions (II) and (III) can be easily satisfied by design of the network.

Assumptions (IV) and (V) only depend on the top-level kernel $K$ and are easy to satisfy in practice. In particular, they always hold for a smooth translation-invariant kernel, such as the Gaussian, as well as the linear kernel.

We are now ready to prove Theorem 1.

**Theorem 1.** *Under Assumptions (I) to (V),*

$$\mathcal{D}_{\mathrm{SMMD}}^{\mu,\Psi^\kappa,\lambda}(\mathbb{P},\mathbb{Q}) \le \frac{L_K\,\kappa^{L/2}}{\gamma\,\sqrt{d_L}\,\alpha^{L/2}}\,\mathcal{W}(\mathbb{P},\mathbb{Q}),$$

*which implies that if $\mathbb{P}_n \xrightarrow{\mathcal{W}} \mathbb{P}$, then $\mathcal{D}_{\mathrm{SMMD}}^{\mu,\Psi^\kappa,\lambda}(\mathbb{P}_n,\mathbb{P}) \to 0$.*

*Proof.* Define the pseudo-distance corresponding to the kernel $k_\psi$

$$d_\psi(x,y) = \|k_\psi(x,\cdot) - k_\psi(y,\cdot)\|_{\mathcal{H}_\psi} = \sqrt{k_\psi(x,x) + k_\psi(y,y) - 2k_\psi(x,y)}.$$

Denote by $\mathcal{W}_{d_\psi}(\mathbb{P},\mathbb{Q})$ the optimal transport metric between $\mathbb{P}$ and $\mathbb{Q}$ using the cost $d_\psi$, given by

$$\mathcal{W}_{d_\psi}(\mathbb{P},\mathbb{Q}) = \inf_{\pi \in \Pi(\mathbb{P},\mathbb{Q})} \mathbb{E}_{(X,Y)\sim\pi}\left[d_\psi(X,Y)\right].$$

where $\Pi$ is the set of couplings with marginals $\mathbb{P}$ and $\mathbb{Q}$. By Lemma 7,

$$\mathrm{MMD}_\psi(\mathbb{P},\mathbb{Q}) \le \mathcal{W}_{d_\psi}(\mathbb{P},\mathbb{Q}).$$

Recall that $\phi_\psi$ is Lipschitz, $\|\phi_\psi\|_{\mathrm{Lip}} < \infty$, so along with Assumption (IV) we have that

$$d_\psi(x,y) \le L_K\|\phi_\psi(x) - \phi_\psi(y)\| \le L_K\|\phi_\psi\|_{\mathrm{Lip}}\|x-y\|.$$

Thus

$$\mathcal{W}_{d_\psi}(\mathbb{P}, \mathbb{Q}) \leq \inf_{\pi \in \Pi(\mathbb{P}, \mathbb{Q})} \mathbb{E}_{(X,Y)\sim\pi}\left[L_K \|\phi_\psi\|_{\mathrm{Lip}} \|X - Y\|\right] = L_K \|\phi_\psi\|_{\mathrm{Lip}} \mathcal{W}(\mathbb{P}, \mathbb{Q}),$$

where $\mathcal{W}$ is the standard Wasserstein distance (2), and so

$$\mathrm{MMD}_\psi(\mathbb{P}, \mathbb{Q}) \leq L_k \|\phi_\psi\|_{\mathrm{Lip}} \mathcal{W}(\mathbb{P}, \mathbb{Q}).$$

We have that $\partial_i \partial_{i+d} k(x, y) = [\partial_i \phi_\psi(x)]^\mathsf{T} \left[ \nabla_a \nabla_b K(a, b)\big|_{(a,b)=(\phi_\psi(x), \phi_\psi(y))} \right] [\partial_i \phi_\psi(y)]$, where the middle term is a $d_L \times d_L$ matrix and the outer terms are vectors of length $d_L$. Thus Assumption (V) implies that $\partial_i \partial_{i+d} k(x, x) \geq \gamma_K^2 \|\partial_i \phi_\psi(x)\|^2$, and hence

$$\sigma_{\mu,k,\lambda}^{-2} \geq \gamma_K^2 \, \mathbb{E}[\|\nabla \phi_\psi(X)\|_F^2]$$

so that

$$\mathrm{SMMD}_\psi^2(\mathbb{P}, \mathbb{Q}) = \sigma_{\mu,k,\lambda}^2 \, \mathrm{MMD}_\psi^2(\mathbb{P}, \mathbb{Q}) \leq \frac{L_K^2 \|\phi_\psi\|_{\mathrm{Lip}}^2}{\gamma_K^2 \, \mathbb{E}\left[\|\nabla \phi_\psi(X)\|_F^2\right]} \, \mathcal{W}^2(\mathbb{P}, \mathbb{Q}).$$

Using Lemma 8, we can write $\phi_\psi(X) = \alpha(\psi) \phi_{\bar\psi}(X)$ with $\bar\psi \in \Psi_1^\kappa$. Then we have

$$\frac{\|\phi_\psi\|_{\mathrm{Lip}}^2}{\mathbb{E}_\mu\left[\|\nabla \phi_\psi(X)\|_F^2\right]} = \frac{\alpha(\psi)^2 \|\phi_{\bar\psi}\|_{\mathrm{Lip}}^2}{\alpha(\psi)^2 \, \mathbb{E}_\mu\left[\|\nabla \phi_{\bar\psi}(X)\|_F^2\right]} \leq \frac{1}{\mathbb{E}_\mu\left[\|\nabla \phi_{\bar\psi}(X)\|_F^2\right]},$$

where we used $\|\phi_{\bar\psi}\|_{\mathrm{Lip}} \leq \prod_{l=1}^L \|\bar{W}^l\| = 1$. But by Lemma 9, for Lebesgue-almost all $X$, $\|\nabla \phi_{\bar\psi}(X)\|_F^2 \geq d_L (\alpha/\kappa)^L$. Using Assumption (I), this implies that

$$\frac{\|\phi_\psi\|_{\mathrm{Lip}}^2}{\mathbb{E}_\mu\left[\|\nabla \phi_\psi(X)\|_F^2\right]} \leq \frac{1}{\mathbb{E}_\mu\left[\|\nabla \phi_{\bar\psi}(X)\|_F^2\right]} \leq \frac{\kappa^L}{d_L \alpha^L}.$$

Thus for any $\psi \in \Psi^\kappa$,

$$\mathrm{SMMD}_\psi(\mathbb{P}, \mathbb{Q}) \leq \frac{L_K \, \kappa^{L/2}}{\gamma_K \sqrt{d_L} \, \alpha^{L/2}} \, \mathcal{W}(\mathbb{P}, \mathbb{Q}).$$

The desired bound on $\mathcal{D}_{\mathrm{SMMD}}^{\mu, \Psi^\kappa, \lambda}$ follows immediately. $\qquad\square$

**Lemma 7.** *Let $(x, y) \mapsto k(x, y)$ be the continuous kernel of an RKHS $\mathcal{H}$ defined on a Polish space $\mathcal{X}$, and define the corresponding pseudo-distance $d_k(x, y) := \|k(x, \cdot) - k(y, \cdot)\|_{\mathcal{H}}$. Then the following inequality holds for any distributions $\mathbb{P}$ and $\mathbb{Q}$ on $\mathcal{X}$, including when the quantities are infinite:*

$$\mathrm{MMD}_k(\mathbb{P}, \mathbb{Q}) \leq \mathcal{W}_{d_k}(\mathbb{P}, \mathbb{Q}).$$

*Proof.* Let $\mathbb{P}$ and $\mathbb{Q}$ be two probability distributions, and let $\Pi(\mathbb{P}, \mathbb{Q})$ be the set of couplings between them. Let $\pi^* \in \arg\min_{(X,Y)\sim\pi}[c_k(X, Y)]$ be an optimal coupling, which is guaranteed to exist [51, Theorem 4.1]; by definition $\mathcal{W}_{d_k}(\mathbb{P}, \mathbb{Q}) = \mathbb{E}_{(X,Y)\sim\pi^*}[d_k(X, Y)]$. When $\mathcal{W}_{d_k}(\mathbb{P}, \mathbb{Q}) = \infty$ the inequality trivially holds, so assume that $\mathcal{W}_{d_k}(\mathbb{P}, \mathbb{Q}) < \infty$.

Take a sample $(X, Y) \sim \pi^\star$ and a function $f \in \mathcal{H}$ with $\|f\|_{\mathcal{H}} \leq 1$. By the Cauchy-Schwarz inequality,

$$\|f(X) - f(Y)\| \leq \|f\|_{\mathcal{H}} \|k(X, \cdot) - k(Y, \cdot)\|_{\mathcal{H}} \leq \|k(X, \cdot) - k(Y, \cdot)\|_{\mathcal{H}}.$$

Taking the expectation with respect to $\pi^\star$, we obtain

$$\mathbb{E}_{\pi^\star}[|f(X) - f(Y)|] \leq \mathbb{E}_{\pi^\star}[\|k(X, \cdot) - k(Y, \cdot)\|_{\mathcal{H}}].$$

The right-hand side is just the definition of $\mathcal{W}_{d_k}(\mathbb{P}, \mathbb{Q})$. By Jensen's inequality, the left-hand side is lower-bounded by

$$|\mathbb{E}_{\pi^\star}[f(X) - f(Y)]| = |\mathbb{E}_{X\sim\mathbb{P}}[f(X)] - \mathbb{E}_{Y\sim\mathbb{Q}}[f(Y)]|$$

since $\pi^\star$ has marginals $\mathbb{P}$ and $\mathbb{Q}$. We have shown so far that for any $f \in \mathcal{H}$ with $\|f\|_{\mathcal{H}} \leq 1$,

$$|\mathbb{E}_\mathbb{P}[f(X)] - \mathbb{E}_\mathbb{Q}[f(Y)]| \leq \mathcal{W}_{c_k}(\mathbb{P}, \mathbb{Q});$$

the result follows by taking the supremum over $f$. $\qquad\square$

**Lemma 8.** *Let $\psi = ((W^L, b^L), (W^{L-1}, b^{L-1}), \ldots, (W^1, b^1)) \in \Psi^\kappa$. There exists a corresponding scalar $\alpha(\psi)$ and $\bar{\psi} = ((\bar{W}^L, \bar{b}^L), (\bar{W}^{L-1}, \bar{b}^{L-1}), \ldots, (\bar{W}^1, \bar{b}^1)) \in \Psi_1^\kappa$, defined by (18), such that for all $X$,*

$$\phi_\psi(X) = \alpha(\psi)\, \phi_{\bar{\psi}}(X).$$

*Proof.* Set $\bar{W}^l = \frac{1}{\|W^l\|} W^l$, $\bar{b}^l = \frac{1}{\prod_{m=1}^l \|W^m\|} b^l$, and $\alpha(\psi) = \prod_{l=1}^L \|W^l\|$. Note that the condition number is unchanged, $\mathrm{cond}(\bar{W}^l) = \mathrm{cond}(W^l) \le \kappa$, and $\|\bar{W}^l\| = 1$, so $\bar{\psi} \in \Phi_1^\kappa$. It is also easy to see from (16) that

$$h_{\bar{\psi}}^l(X) = \frac{1}{\prod_{m=1}^l \|W^m\|} h_\psi^l(X)$$

so that

$$\alpha(\psi) h_{\bar{\psi}}^L(X) = \frac{\prod_{l=1}^L \|W^l\|}{\prod_{l=1}^L \|W^l\|} h_\psi^L(X) = \phi_\psi(X). \qquad \square$$

**Lemma 9.** *Make Assumptions (II) and (III), and let $\psi \in \Psi_1^\kappa$. Then the set of inputs for which any intermediate activation is exactly zero,*

$$\mathcal{N}_\psi = \bigcup_{l=1}^L \bigcup_{k=1}^{d_l} \left\{ X \in \mathbb{R}^d \mid \left( h_\psi^l(X) \right)_k = 0 \right\},$$

*has zero Lebesgue measure. Moreover, for any $X \notin \mathcal{N}_\psi$, $\nabla_X \phi_\psi(X)$ exists and*

$$\|\nabla_X \phi_\psi(X)\|_F^2 \ge \frac{d_L \alpha^L}{\kappa^L}.$$

*Proof.* First, note that the network representation at layer $l$ is piecewise affine. Specifically, define $M_X^l \in \mathbb{R}^{d_l}$ by, using Assumption (III),

$$(M_X^l)_k = \sigma_l'(h_k^l(X)) = \begin{cases} 1 & h_k^l(X) > 0 \\ \alpha & h_k^l(X) < 0 \end{cases};$$

it is undefined when any $h_k^l(X) = 0$, i.e. when $X \in \mathcal{N}_\psi$. Let $V_X^l := W^l \,\mathrm{diag}\left( M_X^{l-1} \right)$. Then

$$h_\psi^l(X) = W^l \sigma_{l-1}(h_\psi^{l-1}(X)) + b^l = V_X^l X + b^l,$$

and thus

$$h_\psi^l(X) = \underline{W}_X^l X + \underline{b}_X^l, \tag{20}$$

where $\underline{b}_X^0 = 0$, $\underline{b}_X^l = V_X^l \underline{b}^{l-1} + b^l$, and $\underline{W}_X^l = V_X^l V_X^{l-1} \cdots V_X^1$, so long as $X \notin \mathcal{N}_\psi$.

Because $\psi \in \Psi_1^\kappa$, we have $\|W^l\| = 1$ and $\sigma_{\min}(W^l) \ge 1/\kappa$; also, $\|M_X^l\| \le 1$, $\sigma_{\min}(M_X^l) \ge \alpha$. Thus $\|\underline{W}_X^l\| \le 1$, and using Assumption (II) with Lemma 10 gives $\sigma_{\min}(\underline{W}_X^l) \ge (\alpha/\kappa)^l$. In particular, each $\underline{W}_X^l$ is full-rank.

Next, note that $\underline{b}_X^l$ and $\underline{W}_X^l$ each only depend on $X$ through the activation patterns $M_X^l$. Letting $H_X^l = (M_X^l, M_X^{l-1}, \ldots, M_X^1)$ denote the full activation patterns up to level $l$, we can thus write

$$h_\psi^l(X) = \underline{W}^{H_X^l} X + \underline{b}^{H_X^l}.$$

There are only finitely many possible values for $H_X^l$; we denote the set of such values as $\mathcal{H}^l$. Then we have that

$$\mathcal{N}_\psi \subseteq \bigcup_{l=0}^L \bigcup_{k=1}^{d_L} \bigcup_{H^l \in \mathcal{H}^l} \left\{ X \in \mathbb{R}^d \mid \underline{W}_k^{H^l} X + \underline{b}_k^{H^l} = 0 \right\}.$$

Because each $\underline{W}_k^{H^l}$ is of rank $d_l$, each set in the union is either empty or an affine subspace of dimension $d - d_l$. As each $d_l > 0$, each set in the finite union has zero Lebesgue measure, and $\mathcal{N}_\psi$ also has zero Lebesgue measure.

We will now show that the activation patterns are piecewise constant, so that $\nabla_X h_\psi^l(X) = \underline{\mathbf{W}}^{H_X^l}$ for all $X \notin \mathcal{N}_\psi$. Because $\psi \in \Psi_1^\kappa$, we have $\|h_\psi^l\|_{\text{Lip}} \leq 1$, and in particular

$$\left| \left( h_\psi^l(X) \right)_k - \left( h_\psi^l(X') \right)_k \right| \leq \|X - X'\|.$$

Thus, take some $X \notin \mathcal{N}_\psi$, and find the smallest absolute value of its activations, $\epsilon = \min_{l=1,\ldots,L} \min_{k=1,\ldots,d_l} \left| \left( h_\psi^l(X) \right)_k \right|$; clearly $\epsilon > 0$. For any $X'$ with $\|X - X'\| < \epsilon$, we know that for all $l$ and $k$,

$$\text{sign} \left( \left( h_\psi^l(X) \right)_k \right) = \text{sign} \left( \left( h_\psi^l(X') \right)_k \right),$$

implying that $H_X^l = H_{X'}^l$ as well as $X' \notin \mathcal{N}_\psi$. Thus for any point $X \notin \mathcal{N}_\psi$, $\nabla \phi_\psi(X) = \underline{\mathbf{W}}^{H_X^L}$. Finally, we obtain

$$\|\nabla \phi_\psi(X)\|_F^2 = \|\underline{\mathbf{W}}^{H_X^L}\|_F^2 \geq d_L \, \sigma_{\min} \left( \underline{\mathbf{W}}^{H_X^L} \right)^2 \geq \frac{d_L \alpha^L}{\kappa^L}. \qquad \square$$

**Lemma 10.** *Let $A \in \mathbb{R}^{m \times n}$, $B \in \mathbb{R}^{n \times p}$, with $m \geq n \geq p$. Then $\sigma_{\min}(AB) \geq \sigma_{\min}(A) \, \sigma_{\min}(B)$.*

*Proof.* A more general version of this result can be found in [19, Theorem 2]; we provide a proof here for completeness.

If $B$ has a nontrivial null space, $\sigma_{\min}(B) = 0$ and the inequality holds. Otherwise, let $\mathbb{R}_*^n$ denote $\mathbb{R}^n \setminus \{0\}$. Recall that for $C \in \mathbb{R}^{m \times n}$ with $m \geq n$,

$$\sigma_{\min}(C) = \sqrt{\lambda_{\min}(C^\mathsf{T} C)} = \sqrt{\inf_{x \in \mathbb{R}_*^n} \frac{x^\mathsf{T} C^\mathsf{T} C x}{x^\mathsf{T} x}} = \inf_{x \in \mathbb{R}_*^n} \frac{\|Cx\|}{\|x\|}.$$

Thus, as $Bx \neq 0$ for $x \neq 0$,

$$\begin{aligned}
\sigma_{\min}(AB) &= \inf_{x \in \mathbb{R}_*^p} \frac{\|ABx\|}{\|x\|} = \inf_{x \in \mathbb{R}_*^p} \frac{\|ABx\|\|Bx\|}{\|Bx\|\|x\|} \\
&\geq \left( \inf_{x \in \mathbb{R}_*^p} \frac{\|ABx\|}{\|Bx\|} \right) \left( \inf_{x \in \mathbb{R}_*^p} \frac{\|Bx\|}{\|x\|} \right) \\
&\geq \left( \inf_{y \in \mathbb{R}_*^n} \frac{\|Ay\|}{\|y\|} \right) \left( \inf_{x \in \mathbb{R}_*^p} \frac{\|Bx\|}{\|x\|} \right) = \sigma_{\min}(A) \, \sigma_{\min}(B). \qquad \square
\end{aligned}$$

### A.2.1 When some of the assumptions don't hold

Here we analyze through simple examples what happens when the condition number can be unbounded, and when Assumption (II), about decreasing widths of the network, is violated.

**Condition Number:** We start by a first example where the condition number can be arbitrarily high. We consider a two-layer network on $\mathbb{R}^2$, defined by

$$\phi_\alpha(X) = \begin{bmatrix} 1 & -1 \end{bmatrix} \sigma(W_\alpha X) \qquad W_\alpha = \begin{bmatrix} 1 & 1 \\ 1 & 1 + \alpha \end{bmatrix} \tag{21}$$

where $\alpha > 0$. As $\alpha$ approaches $0$ the matrix $W_\alpha$ becomes singular which means that its condition number blows up. We are interested in analyzing the behavior of the Lipschitz constant of $\phi$ and the expected squared norm of its gradient under $\mu$ as $\alpha$ approaches $0$.

One can easily compute the squared norm of the gradient of $\phi$ which is given by

$$\|\nabla \phi_\alpha(X)\|^2 = \begin{cases} \alpha^2 & X \in A_1 \\ \gamma^2 \alpha^2 & X \in A_2 \\ (1-\gamma)^2 + (1+\alpha-\gamma)^2 & X \in A_3 \\ (1-\gamma)^2 + (\gamma\alpha + \gamma - 1)^2 & X \in A_4 \end{cases} \tag{22}$$

Here $A_1$, $A_2$, $A_3$ and $A_4$ are defined by (23) and are represented in Figure 4:

$$
\begin{aligned}
A_1 &:= \{X \in \mathbb{R}^2 | X_1 + X_2 \geq 0 \quad X_1 + (1 + \alpha)X_2 \geq 0\} \\
A_2 &:= \{X \in \mathbb{R}^2 | X_1 + X_2 < 0 \quad X_1 + (1 + \alpha)X_2 < 0\} \\
A_3 &:= \{X \in \mathbb{R}^2 | X_1 + X_2 < 0 \quad X_1 + (1 + \alpha)X_2 \geq 0\} \\
A_4 &:= \{X \in \mathbb{R}^2 | X_1 + X_2 \geq 0 \quad X_1 + (1 + \alpha)X_2 < 0\}
\end{aligned}
\tag{23}
$$

Figure 4: Decomposition of $\mathbb{R}^2$ into 4 regions $A_1$, $A_2$, $A_3$ and $A_4$ as defined in (23). As $\alpha$ approaches 0, the area of sets $A_3$ and $A_4$ becomes negligible.

It is easy to see that whenever $\mu$ has a density, the probability of the sets $A_3$ and $A_4$ goes to 0 are $\alpha \to 0$. Hence one can deduce that $\mathbb{E}_\mu[\|\nabla\phi_\alpha(X)\|^2] \to 0$ when $\alpha \to 0$. On the other hand, the squared Lipschitz constant of $\phi$ is given by $(1 - \gamma)^2 + (1 + \alpha - \gamma)^2$ which converges to $2(1 - \gamma)^2$. This shows that controlling the expectation of the gradient doesn't allow to effectively control the Lipschitz constant of $\phi$.

**Monotonicity of the dimensions:** We would like to consider a second example where Assumption (II) doesn't hold. Consider the following two layer network defined by:

$$
\phi(X) = [-1 \quad 0 \quad 1]\, \sigma(W_\beta X) \qquad W_\beta := \begin{bmatrix} 1 & 0 \\ 0 & 1 \\ 1 & \beta \end{bmatrix}
\tag{24}
$$

for $\beta > 0$. Note that $W_\beta$ is a full rank matrix, but Assumption (II) doesn't hold. Depending on the sign of the components of $W_\beta X$ one has the following expression for $\|\nabla\phi_\alpha(X)\|^2$:

$$
\|\nabla\phi_\alpha(X)\|^2 = \begin{cases}
\beta^2 & X \in B_1 \\
\gamma^2\beta^2 & X \in B_2 \\
\beta^2 & X \in B_3 \\
(1 - \gamma)^2 + \gamma^2\beta^2 & X \in B_4 \\
(1 - \gamma)^2 + \beta^2 & X \in B_5 \\
\gamma^2\beta^2 & X \in B_6
\end{cases}
\tag{25}
$$

where $(B_i)_{1 \leq i \leq 6}$ are defined by (26)

$$
\begin{aligned}
B_1 &:= \{X \in \mathbb{R}^2 | X_1 \geq 0 \quad X_2 \geq 0\} \\
B_2 &:= \{X \in \mathbb{R}^2 | X_1 < 0 \quad X_2 < 0\} \\
B_3 &:= \{X \in \mathbb{R}^2 | X_1 \geq \quad X_2 < 0 \quad X_1 + \beta X_2 \geq 0\} \\
B_4 &:= \{X \in \mathbb{R}^2 | X_1 \geq \quad X_2 < 0 \quad X_1 + \beta X_2 < 0\} \\
B_5 &:= \{X \in \mathbb{R}^2 | X_1 > 0 \quad X_2 \geq 0 \quad X_1 + \beta X_2 \geq 0\} \\
B_6 &:= \{X \in \mathbb{R}^2 | X_1 > 0 \quad X_2 \geq 0 \quad X_1 + \beta X_2 < 0\}
\end{aligned}
\tag{26}
$$

The squared Lipschitz constant is given by $\|\phi\|_L^2(1-\gamma)^2 + \beta^2$ while the expected squared norm of the gradient of $\phi$ is given by:

$$\mathbb{E}_\mu[\|\phi(X)\|^2] = 3\beta^2(p(B_1 \cup B_3 \cup B_5) + \gamma^2 p(B_2 \cup B_4 \cup B_6)) + (1-\gamma)^2 p(B_4 \cup B_5). \quad (27)$$

Again the set $B_4 \cup B_5$ becomes negligible as $\beta$ approaches 0 which implies that $\mathbb{E}_\mu[\|\phi(X)\|^2] \to 0$. On the other hand $\|\phi\|_L^2$ converges to $(1-\gamma)^2$. Note that unlike in the first example in (21), the matrix $W_\beta$ has a bounded condition number. In this example, the columns of $W_0$ are all in the null space of $\begin{bmatrix} -1 & 0 & 1 \end{bmatrix}$, which implies $\nabla\phi_0(X) = 0$ for all $X \in \mathbb{R}^2$, even though all matrices have full rank.

# B  DiracGAN vector fields for more losses

Figure 5: Vector fields for different losses with respect to the generator parameter $\theta$ and the feature representation parameter $\psi$; the losses use a Gaussian kernel, and are shown in (28). Following [30], $\mathbb{P} = \delta_0$, $\mathbb{Q} = \delta_\theta$ and $\phi_\psi(x) = \psi x$. The curves show the result of taking simultaneous gradient steps in $(\theta, \psi)$ beginning from three initial parameter values.

Figure 5 shows parameter vector fields, like those in Figure 6, for Example 1 for a variety of different losses:

$$
\begin{aligned}
\text{MMD:} & \quad -\text{MMD}_\psi^2 \\
\text{MMD-GP:} & \quad -\text{MMD}_\psi^2 + \lambda\,\mathbb{E}_\mathbb{P}[(\|\nabla f(X)\| - 1)^2] \\
\text{MMD-GP-Unif:} & \quad -\text{MMD}_\psi^2 + \lambda\,\mathbb{E}_{\widetilde{X} \simeq \mu^*}[(\|\nabla f(\widetilde{X})\| - 1)^2] \\
\text{SN-MMD:} & \quad -2\,\text{MMD}_1(\mathbb{P}, \mathbb{Q})^2 \\
\text{Sobolev-MMD:} & \quad -\text{MMD}_\psi^2 + \lambda(\mathbb{E}_{(\mathbb{P}+\mathbb{Q})/2}[\|\nabla f(X)\|^2] - 1)^2 \qquad (28) \\
\text{CenteredSobolev-MMD:} & \quad -\text{MMD}_\psi^2 + \lambda(\mathbb{E}_{(\mathbb{P}+\mathbb{Q})/2}[\|\nabla f(X)\|^2])^2 \\
\text{LipMMD:} & \quad -\text{LipMMD}_{k_\psi, \lambda}^2 \\
\text{GC-MMD:} & \quad -\text{GCMMD}_{\mathcal{N}(0,10^2), k_\psi, \lambda}^2 \\
\text{SMMD:} & \quad -\text{SMMD}_{k_\psi, \mathbb{P}, \lambda}^2
\end{aligned}
$$

The squared MMD between $\delta_0$ and $\delta_\theta$ under a Gaussian kernel of bandwidth $1/\psi$ and is given by $2(1 - e^{-\frac{\psi^2\theta^2}{2}})$. MMD-GP-unif uses a gradient penalty as in [7] where each samples from $\mu^*$ is obtained by first sampling $X$ and $Y$ from $\mathbb{P}$ and $\mathbb{Q}$ and then sampling uniformly between $X$ and $Y$. MMD-GP uses the same gradient penalty, but the expectation is taken under $\mathbb{P}$ rather than $\mu^*$. SN-MMD refers to MMD with spectral normalization; here this means that $\psi = 1$. Sobolev-MMD refers to the loss used in [33] with the quadratic penalty only. $\mathrm{GCMMD}_{\mu,k,\lambda}$ is defined by (5), with $\mu = \mathcal{N}(0, 10^2)$.

## C   Vector fields of Gradient-Constrained MMD and Sobolev GAN critics

Mroueh et al. [33] argue that *the gradient of the critic (...) defines a transportation plan for moving the distribution mass* (from generated to reference distribution) and present the solution of Sobolev PDE for 2-dimensional Gaussians. We observed that in this simple example the gradient of the Sobolev critic can be very high outside of the areas of high density, which is not the case with the Gradient-Constrained MMD. Figure 6 presents critic gradients in both cases, using $\mu = (\mathbb{P} + \mathbb{Q})/2$ for both.

(a) Gradient-Constrained MMD critic gradient.          (b) Sobolev IPM critic gradient.

Figure 6: Vector fields of critic gradients between two Gaussians. The grey arrows show normalized gradients, i.e. gradient directions, while the black ones are the actual gradients. Note that for the Sobolev critic, gradients norms are orders of magnitudes higher on the right hand side of the plot than in the areas of high density of the given distributions.

This unintuitive behavior is most likely related to the vanishing boundary condition, assummed by Sobolev GAN. Solving the actual Sobolev PDE, we found that the Sobolev critic has very high gradients close to the boundary in order to match the condition; moreover, these gradients point in opposite directions to the target distribution.

## D   An estimator for Lipschitz MMD

We now describe briefly how to estimate the Lipschitz MMD in low dimensions. Recall that

$$\mathrm{LipMMD}_{k,\lambda}(\mathbb{P}, \mathbb{Q}) = \sup_{f \in \mathcal{H}_k : \|f\|^2_{\mathrm{Lip}} + \lambda\|f\|^2_{\mathcal{H}_k} \leq 1} \mathbb{E}_{X \sim \mathbb{P}}[f(X)] - \mathbb{E}_{X \sim \mathbb{Q}}[f(Y)].$$

For $f \in \mathcal{H}_k$, it is the case that

$$\|f\|^2_{\mathrm{Lip}} = \sup_{x \in \mathbb{R}^d} \|\nabla f(x)\|^2 = \sup_{x \in \mathbb{R}^d} \sum_{i=1}^{d} \langle \partial_i k(x, \cdot), f \rangle^2_{\mathcal{H}_k} = \sup_{x \in \mathbb{R}^d} \left\langle f, \sum_{i=1}^{d} [\partial_i k(x, \cdot) \otimes \partial_i k(x, \cdot)] f \right\rangle_{\mathcal{H}_k}.$$

Thus we can approximate the constraint $\|f\|^2_{\mathrm{Lip}} + \lambda\|f\|^2_{\mathcal{H}_k} \leq 1$ by enforcing the constraint on a set of $m$ points $\{Z_i\}$ reasonably densely covering the region around the supports of $\mathbb{P}$ and $\mathbb{Q}$, rather

than enforcing it at every point in $\mathcal{X}$. An estimator of the Lipschitz MMD based on $X \sim \mathbb{P}^{n_X}$ and $Y \sim \mathbb{Q}^{n_Y}$ is

$$\widehat{\text{LipMMD}}_{k,\lambda}(X, Y, Z) \approx \sup_{f \in \mathcal{H}_k} \frac{1}{n_X} \sum_{j=1}^{n_X} f(X_j) - \frac{1}{n_Y} \sum_{j=1}^{n_Y} f(Y_j) \tag{29}$$

$$\text{s.t. } \forall j, \ \|\nabla f(Z_j)\|^2 + \lambda \|f\|_{\mathcal{H}_k}^2 \le 1.$$

By the generalized represener theorem, the optimal $f$ for (29) will be of the form

$$f(\cdot) = \sum_{j=1}^{n_X} \alpha_j k(X_j, \cdot) + \sum_{j=1}^{n_Y} \beta_j k(Y_j, \cdot) + \sum_{i=1}^{d} \sum_{j=1}^{m} \gamma_{(i,j)} \partial_i k(Z_j, \cdot).$$

Writing $\delta = (\alpha, \beta, \gamma)$, the objective function is linear in $\delta$,

$$\begin{bmatrix} \frac{1}{n_X} & \cdots & \frac{1}{n_X} & -\frac{1}{n_Y} & \cdots & -\frac{1}{n_Y} & 0 & \cdots & 0 \end{bmatrix} \delta.$$

The constraints are quadratic, built from the following matrices, where the $X$ and $Y$ samples are concatenated together, as are the derivatives with each dimension of the $Z$ samples:

$$K := \begin{bmatrix} k(X_1, X_1) & \cdots & k(X_1, Y_{n_Y}) \\ \vdots & \ddots & \vdots \\ k(Y_{n_X}, X_1) & \cdots & k(Y_{n_Y}, Y_{n_Y}) \end{bmatrix}$$

$$B := \begin{bmatrix} \partial_1 k(Z_1, X_1) & \cdots & \partial_1 k(Z_1, Y_{n_Y}) \\ \vdots & \ddots & \vdots \\ \partial_d k(Z_m, X_1) & \cdots & \partial_d k(Z_m, Y_{n_Y}) \end{bmatrix}$$

$$H := \begin{bmatrix} \partial_1 \partial_{1+d} k(Z_1, Z_1) & \cdots & \partial_1 \partial_{d+d} k(Z_1, Z_m) \\ \vdots & \ddots & \vdots \\ \partial_d \partial_{1+d} k(Z_m, Z_1) & \cdots & \partial_d \partial_{d+d} k(Z_m, Z_m) \end{bmatrix}.$$

Given these matrices, and letting $O_j = \sum_{i=1}^{d} e_{(i,j)} e_{(i,j)}^{\mathsf{T}}$ where $e_{(i,j)}$ is the $(i,j)$th standard basis vector in $\mathbb{R}^{md}$, we have that

$$\|f\|_{\mathcal{H}_k}^2 = \delta^{\mathsf{T}} \begin{bmatrix} K & B^{\mathsf{T}} \\ B & H \end{bmatrix} \delta \qquad \|\nabla f(Z_j)\|^2 = \sum_{i=1}^{d} (\partial_i f(Z_j))^2 = \delta^{\mathsf{T}} \begin{bmatrix} B^{\mathsf{T}} O_j B & B^{\mathsf{T}} O_j H \\ H O_j B & H O_j H \end{bmatrix} \delta.$$

Thus the optimization problem (29) is a linear problem with convex quadratic constraints, which can be solved by standard convex optimization software. The approximation is reasonable only if we can effectively cover the region of interest with densely spaced $\{Z_i\}$; it requires a nontrivial amount of computation even for the very simple 1-dimensional toy problem of Example 1.

One advantage of this estimator, though, is that finding its derivative with respect to the input points or the kernel parameterization is almost free once we have computed the estimate, as long as our solver has computed the dual variables $\mu$ corresponding to the constraints in (29). We just need to exploit the envelope theorem and then differentiate the KKT conditions, as done for instance in [1]. The differential of (29) ends up being, assuming the optimum of (29) is at $\hat{\delta} \in \mathbb{R}^{n_X + n_Y + md}$ and $\hat{\mu} \in \mathbb{R}^m$,

$$\text{d}\widehat{\text{LipMMD}}_{k,\lambda}(X, Y, Z) = \hat{\delta}^{\mathsf{T}} \begin{bmatrix} \text{d}K \\ \text{d}B \end{bmatrix} \begin{bmatrix} \frac{1}{n_X} & \cdots & \frac{1}{n_X} & -\frac{1}{n_Y} & \cdots & -\frac{1}{n_Y} \end{bmatrix}^{\mathsf{T}} - \sum_{j=1}^{m} \hat{\mu}_j \hat{\delta}^{\mathsf{T}}(\text{d}P_j)\hat{\delta}$$

$$\text{d}P_j := \begin{bmatrix} (\text{d}B)^{\mathsf{T}} O_j B + B^{\mathsf{T}} O_j(\text{d}H) & (\text{d}B)^{\mathsf{T}} O_j H + B^{\mathsf{T}} O_j(\text{d}H) \\ (\text{d}H)O_j B + H O_j(\text{d}B) & (\text{d}H)O_j H + H O_j(\text{d}H) \end{bmatrix} + \lambda \begin{bmatrix} \text{d}K & \text{d}B^{\mathsf{T}} \\ \text{d}B & \text{d}H \end{bmatrix}.$$

# E    Near-equivalence of WGAN and linear-kernel MMD GANs

For an MMD GAN-GP with kernel $k(x, y) = \phi(x)\phi(y)$, we have that

$$\text{MMD}_k(\mathbb{P}, \mathbb{Q}) = |\mathbb{E}_{\mathbb{P}} \phi(x) - \mathbb{E}_{\mathbb{Q}} \phi(Y)|$$

and the corresponding critic function is

$$\frac{\eta(t)}{\|\eta\|_{\mathcal{H}}} = \frac{\mathbb{E}_{X \sim \mathbb{P}} \phi(X)\phi(t) - \mathbb{E}_{Y \sim \mathbb{Q}} \phi(Y)\phi(t)}{|\mathbb{E}_{\mathbb{P}} \phi(X) - \mathbb{E}_{\mathbb{Q}} \phi(Y)|} = \operatorname{sign} \left( \mathbb{E}_{X \sim \mathbb{P}} \phi(X) - \mathbb{E}_{Y \sim \mathbb{Q}} \phi(Y) \right) \phi(t).$$

Thus if we assume $\mathbb{E}_{X \sim \mathbb{P}} \phi(X) > \mathbb{E}_{Y \sim \mathbb{Q}} \phi(Y)$, as that is the goal of our critic training, we see that the MMD becomes identical to the WGAN loss, and the gradient penalty is applied to the same function.

(MMD GANs, however, would typically train on the unbiased estimator of $\mathrm{MMD}^2$, giving a very slightly different loss function. [7] also applied the gradient penalty to $\eta$ rather than the true critic $\eta/\|\eta\|$.)

The SMMD with a linear kernel is thus analogous to applying the scaling operator to a WGAN; hence the name SWGAN.

# F   Additional experiments

## F.1   Comparison of Gradient-Constrained MMD to Scaled MMD

Figure 7 shows the behavior of the MMD, the Gradient-Constrained SMMD, and the Scaled MMD when comparing Gaussian distributions. We can see that $\mathrm{MMD} \propto \mathrm{SMMD}$ and the Gradient-Constrained MMD behave similarly in this case, and that optimizing the $\mathrm{SMMD}$ and the Gradient-Constrained MMD is also similar. Optimizing the MMD would yield an essentially constant distance.

## F.2   IGMs with Optimized Gradient-Constrained MMD loss

We implemented the estimator of Proposition 3 using the empirical mean estimator of $\eta$, and sharing samples for $\mu = \mathbb{P}$. To handle the large but approximately low-rank matrix system, we used an incomplete Cholesky decomposition [43, Algorithm 5.12] to obtain $R \in \mathbb{R}^{\ell \times M(1+d)}$ such that $\begin{bmatrix} K & G^{\mathsf{T}} \\ G & H \end{bmatrix} \approx R^{\mathsf{T}} R$. Then the Woodbury matrix identity allows an efficient evaluation:

$$\left( R^{\mathsf{T}} R + M\lambda I \right)^{-1} = \frac{1}{M\lambda} \left( I - R(RR^{\mathsf{T}} + M\lambda I)^{-1} R \right).$$

Even though only a small $\ell$ is required for a good approximation, and the full matrices $K$, $G$, and $H$ need never be constructed, backpropagation through this procedure is slow and not especially GPU-friendly; training on CPU was faster. Thus we were only able to run the estimator on MNIST, and even that took days to conduct the optimization on powerful workstations.

The learned models, however, were reasonable. Using a DCGAN architecture, batches of size 64, and a procedure that otherwise agreed with the setup of Section 4, samples with and without spectral normalization are shown in Figures 8a and 8b. After the points in training shown, however, the same rank collapse as discussed in Section 4 occurred. Here it seems that spectral normalization may have delayed the collapse, but not prevented it. Figure 8c shows generator loss estimates through training, including the obvious peak at collapse; Figure 8d shows KID scores based on the MNIST-trained convnet representation [7], including comparable SMMD models for context. The fact that SMMD models converged somewhat faster than Gradient-Constrained MMD models here may be more related to properties of the estimator of Proposition 3 rather than the distances; more work would be needed to fully compare the behavior of the two distances.

## F.3   Spectral normalization and Scaled MMD

Figure 9 shows the distribution of critic weight singular values, like Figure 2, at more layers. Figure 11 and Table 2 show results for the spectral normalization variants considered in the experiments. MMDGAN, with neither spectral normalization nor a gradient penalty, did surprisingly well in this case, though it fails badly in other situations.

Figure 9 compares the decay of singular values for layer of the critic's network at both early and later stages of training in two cases: with or without the spectral parametrization. The model was trained on CelebA using SMMD. Figure 11 shows the evolution per iteration of Inception score,

Figure 7: Plots of various distances between one dimensional Gaussians, where $P = \mathcal{N}(0, 0.1^2)$, and the colors show $\log \mathcal{D}(P, \mathcal{N}(\mu, \sigma^2))$. All distances use $\lambda = 1$. Top left: MMD with a Gaussian kernel of bandwidth $\psi = 0.1$. Top right: MMD with bandwidth $\psi = 10$. Middle left: Gradient-Constrained MMD with bandwidth $\psi = 0.1$. Middle right: Gradient-Constrained MMD with bandwidth $\psi = 10$. Bottom left: Optimized SSMD, allowing any $\psi \in \mathbb{R}$. Bottom right: Optimized Gradient-Constrained MMD.

(a) Without spectral normalization; 32 000 generator iterations.

(b) With spectral normalization; 41 000 generator iterations.

(c) Generator losses.

(d) KID scores.

Figure 8: The MNIST models with Optimized Gradient-Constrained MMD loss.

FID and KID for Sobolev-GAN, MMDGAN and variants of MMDGAN and WGAN using spectral normalization. It is often the case that this parametrization alone is not enough to achieve good results.

Figure 9: Singular values at different layers, for the same setup as Figure 2.

## F.4 Additional samples

Figures 12 and 13 give extra samples from the models.

Figure 10: Evolution of various quantities per generator iteration on CelebA during training. 4 models are considered: (SMMDGAN, SN-SMMDGAN, MMDGAN, SN-MMDGAN). (a) Loss: $\text{SMMD}^2 = \sigma^2_{\mu,k,\lambda}\,\text{MMD}^2_k$ for SMMDGAN and SN-SMMDGAN, and $\text{MMD}^2_k$ for MMDGAN and SN-MMDGAN. The loss saturates for MMDGAN (green); spectral normalization allows some improvement in loss, but training is still unstable (orange). SMMDGAN and SN-SMMDGAN both lead to stable, fast training (blue and red). (b) SMMD controls the critic complexity well, as expected (blue and red); SN has little effect on the complexity (orange). (c) Ratio of the highest singular value to the smallest for the first layer of the critic network: $\sigma_{\max}/\sigma_{\min}$. SMMD tends to increase the condition number of the weights during training (blue), while SN helps controlling it (red). (d) KID score during training: Only variants using SMMD lead to stable training in this case.

Figure 11: Evolution per iteration of different scores for variants of methods, mostly using spectral normalization, on CIFAR-10.

Table 2: Mean (standard deviation) of score evaluations on CIFAR-10 for different methods using Spectral Normalization.

| Method | IS | FID | KID$\times 10^3$ |
|---|---|---|---|
| MMDGAN | 5.5±0.0 | 73.9±0.1 | 39.4±1.5 |
| SN-WGAN | 2.2±0.0 | 208.5±0.2 | 178.9±1.5 |
| SN-WGAN-GP | 2.5±0.0 | 154.3±0.2 | 125.3±0.9 |
| SN-Sobolev-GAN | 2.9±0.0 | 140.2±0.2 | 130.0±1.9 |
| SN-MMDGAN-GP | 4.6±0.1 | 96.8±0.4 | 59.5±1.4 |
| SN-MMDGAN-L2 | 7.1±0.1 | 31.9±0.2 | 21.7±0.9 |
| SN-MMDGAN | 6.9±0.1 | 31.5±0.2 | 21.7±1.0 |
| SN-MMDGAN-GP-L2 | 6.9±0.2 | 32.3±0.3 | 20.9±1.1 |
| SN-SMMDGAN | **7.3±0.1** | **25.0±0.3** | **16.6±2.0** |

Figure 12: Samples from a generator trained on ImageNet dataset using Scaled MMD with Spectral Normalization: SN-SMMDGAN.

(a) SNGAN

(b) SobolevGAN

(c) MMDGAN-GP-L2

(d) SN-SMMD GAN

(e) SN SWGAN

(f) SMMD GAN

Figure 13: Comparison of samples from different models trained on CelebA with $160 \times 160$ resolution.