[Reviews · NeurIPS 2018]

Reviewer 1



In this paper, the authors propose a method for gradient-based regularization of the critic of MMD GAN-like models. Gradient constrained maximum mean discrepancy has been proposed in the paper. The presentation of the paper is clear however the results present in the section 3 are trivial and does not provide any new insights on understanding GAN. I would also assume that the original proposed approach would take a long time to train and the authors also mention this in the paper. Hence they propose to optimize the lower bound of the Gradient-Constrained MMD instead which essentially turns out to be the scaled version of MMD. But it is not very clear why the scaled MMD is performing better over vanilla MMD. Ideally one should expect similar behaviour for both of them.

Reviewer 2



This paper studied the gradient regularization for MMD-based critics in GANs. It proposed gradient constrained MMD and its approximation the Scaled MMD, which is a principle way to control the gradient. The results on several datasets show the effectiveness of SN-SMMDGAN. The idea of the paper is novel and elegant, solving an important problem for MMD-based GANs. The experimental results are pretty solid, compared will most related state-of-the-arts approaches on CIFAR, CelebA and ImageNet. The only concern is the performance of Sobolev GAN, which had some gap between CIFAR and CelebA. It is better to have more analysis on this and disclose the setting of it. The paper is well written, with good motivation and clear pipeline. It well described the relations among different regularizers (for general GANs and MMD GANs).

Reviewer 3



Summary: The paper presents a gradient constrained MMD that generalizes the Sobolev discrepancy introduced in Sobolev GAN by adding an L_2 regularizer to the gradient regularizer (the kernel case was also discussed in the aforementioned paper). Computing this regularized MMD is computationally expensive authors propose a bound on the this discrepancy , and optimize with the new scaled MMD , that changes the radius of the ball in the RKHS according to the kernel and its derivative statistics on the target distribution in this paper. The paper then combines the Scaled MMD with the recently introduced spectral normalization in order to train GAN. Comments: - Line 88: phi_psi: X to R^s. It seems throughout the paper that s is considered to be equal 1. It would be good to mention this early on. Have you tried s >1? - Line 117 ||f||_{L} if this is the Lipchitz norm this is not correct one need to consider the definition with sup_{x,y} f(x)- f(y)/d(x,y), the one given is a bound. - Line 208 the gaussian kernel is considered always with sigma =1? - Table (2 b) SMMDGAN without SN is not given, it would been nice to have it as well to tease apart the contribution of SN and the scaled MMD. Line 247 and appendix, I think some more ablation study is needed to understand how this scaling and the spectral normalization interact.( see point below on tracking the RKHS norm and singular values with and without SN might help) - Have you tracked the cost function in the training process? How does the numerator and denominator behave? It would be great to see how the radius in the RKHS behave in the training and how this relates to the plots of singular values you give in Figure 2 with SN and w/ SN. - Line 242, on Sobolev GAN it seems that on CIFAR 10 there is an issue in the hyperparameter selection or the implementation or the choice of mu as it is set in this paper to mu =P. In the reviewer experience Sobolev GAN behaves at least close to WGAN-GP on CIFAR 10. Minor: - The figure captions are too sparse, and one needs to jump to the text to know what is going on. It would be good to have better descriptive captions it helps in the readability of the paper. -Figure 6 (a) and (b) are the same - would be good to have on a toy data like the mixture of gaussians to train the GAN with the original cost function given in proposition 4.